# Species-specific temporal variation in photosynthesis as a moderator of peatland carbon sequestration

Aino Korrensalo[1], Pavel Alekseychik[2], Tomáš Hájek[3], Janne Rinne[4], Timo Vesala[2,5], Lauri Mehtätalo[6], Ivan Mammarella[2], Eeva-Stiina Tuittila[1]

[1]School of Forest Sciences, University of Eastern Finland, Joensuu, Finland

[2] Dept. of Physics, University of Helsinki, Helsinki, Finland

[3] Faculty of Science, University of South Bohemia, České Budějovice, Czech Republic

[4] Dept. of Physical Geography and Ecosystem Science, Lund University, Lund, Sweden

[5] Dept. of Forest Sciences, University of Helsinki, Helsinki, Finland

[6] School of Computing, University of Eastern Finland, Joensuu, Finland

*Correspondence to*: aino.korrensalo@uef.fi

**Abstract.** In boreal bogs plant species are low in number, but they differ greatly in their growth forms and photosynthetic properties. We assessed how ecosystem carbon (C) sink dynamics were affected by seasonal variations in photosynthetic rate and leaf area of different species. Photosynthetic properties (light-response parameters), leaf area development and areal cover (abundance) of the species were used to quantify species-specific net and gross photosynthesis rates ($P_N$ and $P_G$, respectively), which were summed to express ecosystem-level $P_N$ and $P_G$. The ecosystem-level $P_G$ was compared with a gross primary production (GPP) estimate derived from eddy covariance measurements (EC).

Species areal cover, rather than differences in photosynthetic properties, determined the species with the highest $P_G$ of both vascular plants and *Sphagna*. Species-specific contributions to the ecosystem $P_G$ varied over the growing season, which, in turn, determined the seasonal variation in ecosystem $P_G$. The upscaled growing-season $P_G$ estimate, 230 g C m$^{-2}$, agreed well with the GPP estimated by the EC, 243 g C m$^{-2}$.

*Sphagna* were superior to vascular plants in ecosystem-level $P_G$ throughout the growing season, but had a lower $P_N$. $P_N$ results indicated that areal cover of the species, together with their differences in photosynthetic parameters, shape the ecosystem-level C balance. Species with low areal cover but high photosynthetic efficiency, appear to be potentially important for the ecosystem C sink. Results imply that functional diversity, i.e. the presence of plant groups with different seasonal timing and efficiency of photosynthesis, may increase the stability of C sink of boreal bogs.

**Key-words**

$CO_2$, ecosystem stability, insurance hypothesis, vascular plant, *Sphagnum*

## 1 Introduction

Northern peatlands are a globally important carbon (C) sink and storage of approximately 500 gigatons of C (Gorham, 1991; Yu et al., 2012) as a result of an imbalance between photosynthesis and decomposition. Boreal bogs are peatland ecosystems where photosynthetic productivity is limited by mid-summer dry periods, light induced stress and, in particular, low nutrient availability (Frolking et al., 1998; Moore et al., 2002). Due to the low rate of photosynthesis, the annual C sink of boreal bogs is weak and sensitive to changes; even a small change in the environmental conditions that regulate the C cycle can turn the ecosystem into a C source (Waddington and Roulet, 2000; Lund et al., 2012). The rate by which $CO_2$ enters the ecosystem through photosynthesis of all of the individual plants together is the definition of gross primary production (GPP). When ecosystem respiration ($R_{eco}$) is subtracted from GPP, the result is net ecosystem exchange (NEE) of $CO_2$ between the ecosystem and the atmosphere. On the scale of individual plants, the same processes are called gross photosynthesis ($P_G$), plant respiration (R) and net photosynthesis ($P_N$), respectively (Chapin et al. 2011).

Boreal bogs are ecosystems with low species diversity but high diversity of growth forms due to the large microtopographical variation and associated diversity of habitats along the water table gradient (Turetsky et al., 2012). Several studies (e.g. Weltzin et al., 2000; Moore et al., 2002; Leppälä et al., 2008) have reported that patterned bogs produce more biomass and have less variation in gross photosynthesis over the growing season than fens, which receive additional nutrients from the surrounding mineral soil and generally have more homogenous, sedge-dominated vegetation (Weltzin et al., 2000). Experimental studies have shown that bog plant growth forms have differential responses to warming and water table level manipulation, which can help to maintain the level of total ecosystem productivity under changing environmental conditions (Weltzin et al., 2000; Breeuwer et al., 2009). Short-term plant removal experiments have shown the differential roles of plant functional types for the peatland NEE and GPP (Ward et al., 2009; Kuiper et al., 2014; Robroek et al., 2015). Photosynthetic properties of bog plants are known to differ widely between species of the same functional type (Small, 1972) and between phases of growing season (Korrensalo et al., 2016). So far, the role of species-level differences in temporal variation of bog ecosystem photosynthesis has not been studied.

Here, we aim to solve the linkage between the temporal pattern of bog carbon balance and the development of species-specific potential photosynthesis and leaf area. For this purpose, we quantified the contribution of different plant species to ecosystem-level photosynthesis over a growing season. As species differ in their photosynthetic properties, and the properties vary over the growing season, we expect their importance for the ecosystem carbon sequestration also vary over the season. To reach our aim we estimate $P_N$ and $P_G$ for the whole study site based on the monthly species-level light response of photosynthesis and species-specific leaf area development over a growing season. To validate the upscaling approach, the sum of species-level $P_G$ is compared to the GPP derived from eddy covariance (EC) measurements at the study site.

## 2 Methods

### 2.1 Study site

The study site (61° 50.179' N, 24° 10.145' E) is situated in an ombrotrophic bog, which is a part of the Siikaneva peatland complex in Southern Finland, located in the southern boreal vegetation zone. The annual temperature

sum in the area (base temperature 5 °C) is 1318 degree days, annual rainfall is 707 mm and the average annual, January and July temperatures are 4.2, -7.2 and 17.1 °C, respectively (30 year averages (years 1982–2011) from Juupajoki-Hyytiälä weather station). The study site has a surface topography typical of raised bogs that varies from open water pools and mud surfaces to hollows, lawns and hummocks. An EC flux tower is mounted on a raft in the center of the site.

The vegetation is mainly composed of 11 vascular plant and eight *Sphagnum* species (Table 1), the abundance of which varies markedly along the microtopographical gradient. A continuous *Sphagnum* carpet covers the surfaces from hummocks to hollows, although no *Sphagna* are present on the mud and water surfaces. *Sphagnum cuspidatum* and *S. majus* are dominant in hollows, *S. papillosum, S. rubellum, S. balticum* and *S. magellanicum* in lawns, and *S. fuscum*, *S. rubellum* and *S. angustifolium* cover the hummocks. Vascular plant species composition includes *Rhynchospora alba*, *Scheuchzeria palustris* and *Carex limosa* vegetation on mud and hollow surfaces, dwarf shrubs (*Andromeda polifolia*, *Vaccinium oxycoccos*) and *Eriophorum vaginatum* on lawn surfaces, and a thick shrub layer of *Calluna vulgaris*, *Betula nana*, *Empetrum nigrum*, and sedges *Eriophorum vaginatum* and *Trichophorum cespitosum* on hummocks.

**2.2 Plant level photosynthesis measurements**

To quantify the role of plant species in ecosystem-level photosynthesis over the growing season, we conducted net photosynthesis ($P_N$) measurements of 19 most common species at the study site. Over the growing season 2013, we measured $CO_2$ exchange of 3–5 samples of each species per month at three light levels with two open, flow-through gas exchange measurement devices (GFS-3000, Walz, Germany and LI-6400, LI-COR, USA). Samples were collected from several locations inside the study area a maximum of 30 hours prior to measurement; *Sphagna* were collected into small plastic bags and vascular plants into plastic boxes with an ample amount of roots and peat. These were kept moist until measured. Vascular plants were kept in shaded conditions and *Sphagna* were stored in the dark at 5 °C. The moss cuvettes were filled with *Sphagnum* capitula imitating their natural shoot density of each species in the field, resulting in a total number of 6–16 capitula inside a cuvette depending on the species. Before placing the capitula into the cuvettes, they were first wetted and then lightly dried of excess water with pulp paper. The cuvette was then placed under a photosynthetic photon flux density (PPFD) of approximately 1000 µmol $m^{-2}$ $s^{-1}$ to acclimate to light for 20 min. The measured light levels were 2000, 25 and 0 µmol $m^{-2}$ $s^{-1}$ for vascular plants and 2000, 35 and 0 µmol $m^{-2}$ $s^{-1}$ for *Sphagna*. Light levels were designed to both catch the linear, light-limited beginning and the light-saturated maximum of the light response curve without causing photoinhibitory reduction of photosynthetic rate (Laine et al., 2015). Vascular plants were measured with a standard cuvette, but for *Sphagna* we used a moss cuvette of our own construction with internal dimensions of 3 × 2 × 1 cm (Hájek et al., 2009) with a net bottom to allow airflow above and below the sample reducing the aerodynamic resistance. After changing the light level, we waited for the $P_N$ to reach steady state before recording the $CO_2$ exchange. Only PPFD was varied during the measurements, while we kept air temperature at constant 20 °C, the flow rate at 600 µmol $s^{-1}$ and the $CO_2$ concentration in the incoming air at 400 ppm to be able to compare the seasonal changes in photosynthetic potential among species. The relative humidity inside the cuvette was kept at 50% for the vascular plants and below 90% for the *Sphagna*. The measured $P_N$ value of each sample at the three light levels was expressed per photosynthesizing leaf area (mg $CO_2$ $m^{-2}$ (LA) $h^{-1}$), which was the leaf area inside the cuvette measured with a scanner for vascular plants and assumed to be the cuvette area for *Sphagna*. Two of

the species, namely *Rhynchospora alba* and *Rubus chamaemorus*, were not yet of measurable size in May; *R. alba* had already mostly senesced in September and therefore were not measured in those months. Altogether, the data consisted of 720 measurements.

**2.3 Net photosynthesis model**

To obtain a species-wise flux reconstruction of $P_N$ and $P_G$, we fitted a nonlinear mixed-effects model separately for each combination of species and month. Mixed-effects modeling approach allowed us to take into account the variation between samples, of which each was measured at three light levels. We used the hyperbolic light saturation curve of $P_N$ (Larcher, 2003):

$$PN_{si} = R_s + \frac{Pmax_s \, PPFDsi}{k_s + PPFDsi} + e_{si} \qquad (1)$$

where $PN_{si}$ is the observed net $CO_2$ exchange (mg $CO_2$ m$^{-2}$ (LA) h$^{-1}$) and $PPFD_{si}$ is the photosynthetic photon flux density for measurement $i$ of sample $s$. The three parameters to be estimated are the maximum rate of light-saturated net photosynthesis ($Pmax_s$), the PPFD level where half of Pmax was reached ($k_s$) and respiration ($R_s$), and were assumed to be constant for each combination of species and month. $e_{si}$ is the normally distributed residual variance of the model with a mean of zero. Normally distributed random effect of the sample was included in one to three of the parameters depending on the model. The random effect structure was selected based on the Akaike information criterion (AIC) values of the alternative models with random effects included in a different combination of parameters. The random effects for the sub-models of each nonlinear model form a vector of random effect with non-zero correlation. Parameter values for the 87 $P_N$ models are presented in Supplementary information (Table S1). The separate fitting for each combination of species and month leads to models with similar asymptotic unbiasedness as a single model for complete data would do. However, separate models do not quantify the temporal and between-species correlation of random effects and residuals, and therefore do not provide a sufficiently detailed model for evaluating the prediction errors of upscaled estimates of net photosynthesis. A proper model for such purpose would model both temporal and between-species covariance of both random effects and residuals, but model fitting procedures for such model are not available in the standard statistical software. All models were fitted using the functions *nlme* of the R program package *nlme* (Pinheiro and Bates, 2000).

*Sphagna* were not measured in June nor were vascular plants in July due to technical failures of the devices. Then, light response curves for these two months were fitted by combining data from the previous and following month for each species. In 5 of the 95 species–month combinations only one sample of the species had an acceptable measurement. The parameters for these months were estimated separately without the mixed model structure (Supplementary information, Table S1).

**2.4 Upscaling**

To upscale species-level photosynthesis to the ecosystem level, the cover of each species was estimated within the study site with a systematic vegetation inventory conducted in July 2013. We estimated the relative cover of each plant species in 121 0.071 m$^2$ plots (Table 1), which were arranged in a regular grid in a 30 m radius circle around the EC tower. To link net photosynthesis measured per leaf area to species cover, we converted species cover in

the study area to leaf area index (LAI) using linear relationships between the two (Supplementary information, Table S2). Relationships were based on an inventory made in July 2012 over a 200 m radius circle where species cover was estimated, and then all living aboveground vegetation was harvested from 65 0.071 $m^2$ inventory plots for LAI measurements. The vascular plant LAI of these samples was measured in the laboratory.

5 We monitored LAI development of each vascular plant species over the growing season in 18 permanent sampling plots (0.36 $m^2$) that represented all the vegetation communities (n=3 in each vegetation community) along the microtopographical gradient in the study site. LAI was estimated every third week according to method described by Wilson et al., (2007). Continuous LAI development of each species was then estimated by fitting a log-linear response to the observations. The shape of the log-linear LAI development was taken from this fitting and the 10 growing season LAI maximum for each species was taken from the converted average cover (Table 1). *Sphagnum* leaf area was assumed to stay constant over the whole growing season and was obtained using the average cover from the 2013 inventory.

Using the light response curves, estimated daily LAI, and half-hourly, above-canopy PPFD data from SMEAR II measurement station (61° 50.845' N, 24° 17.686' E), we calculated $P_N$ and $P_G$ for each half hour period (mg $CO_2$ 15 $m^{-2}$ 30 $min^{-1}$) over the growing season (Julian days 121–273) with the species-wise and monthly light response curves. $P_G$ was calculated with the same model without the R parameter, i.e. assuming that respiration is zero. Model predictions were not meaningfully changed by using marginal prediction, i.e. averaging the predictions over the distribution random effects (e.g. de-Miguel et al., 2012) and were therefore computed using the fixed part of the model only. Growing season $P_N$ and $P_G$ of the whole study site were calculated as a sum of their daily values.

20 **2.5 Ecosystem-level $CO_2$ exchange measurements and estimation of gross primary production**

To validate the measured levels of photosynthesis, the calculated values were compared with the GPP estimates obtained by EC measurements, which offer an independent estimate of the ecosystem-level $CO_2$ exchange measured directly as turbulent vertical fluxes (e.g. Baldocchi, 2003; Aubinet et al., 2012). The EC system comprised a 3-D ultrasonic anemometer (USA-1, METEK Meteorologische Messtechnik GmbH, Germany) and 25 an enclosed $H_2O/CO_2$ gas analyzer (LI-7200, LI-COR Biosciences, USA). The EC sensors were mounted on the mast 2.5 m above the peat surface. EddyUH software was used to process the raw data and produce the 30-min average fluxes of latent heat, sensible heat, and $CO_2$ (Mammarella et al., 2015). Standard EC data checks based on the widely accepted quality criteria (e.g. Aubinet et al., 2012) were applied partly automatically by the software and partly manually; the EC data at friction velocity ($u_*$) less than 0.1 m $s^{-1}$ were rejected. The resulting EC fluxes 30 represent the exchange over a quasi-elliptical source area (footprint) located within about 30 m upwind of the EC mast, as suggested by footprint calculations using the model by Kormann and Meixner, (2001).

NEE measured by the EC method was then partitioned into ecosystem GPP and $R_{eco}$. The daytime $R_{eco}$ estimates were obtained from the $Q_{10}$-type temperature response curve fitted to the nighttime EC data, when respiration is the only component of NEE. Nighttime was defined as all the periods when the sun elevation angle was lower than 35 5°. Peat temperature at 5cm depth was used as the driver of $R_{eco}$, in the following relationship:

$$R_{eco} = R_{ref} Q_{10}^{\left(\frac{T_p - T_{ref}}{10}\right)} \qquad (2)$$

where $T_p$ is the peat temperature at a 5cm depth (°C) and $T_{ref}$ is the peat reference temperature of 12°C. Parameters to be estimated from the fit of the model (Eq. (2)) to all available night time NEE data were $R_{ref}$, the reference respiration at the temperature of 12 °C, and $Q_{10}$, the temperature sensitivity coefficient.

The GPP estimates were calculated by subtracting the modeled $R_{eco}$ from the EC-derived NEE values. Finally, in order to gap-fill the GPP time series, a model using PPFD (from the SMEAR II measurement station) and the footprint-scale LAI was fitted to the data following:

$$GPP = \frac{Pmax \; PPFD}{k + PPFD}(LAI + b) \qquad (3)$$

where GPP is expressed in mg ($CO_2$) m$^{-2}$ h$^{-1}$. LAI is the modeled daily vascular leaf area index described above, while $b$ represents the temporally constant contribution of the *Sphagnum* to total LAI.

One of the major sources of random uncertainty in cumulative GPP originates from the uncertainty in the $R_{eco}$ and GPP model parameters. Random uncertainty was calculated as the 95% confidence interval of a set of 1000 cumulative GPP estimates obtained using $R_{ref}$, $Q_{10}$, $P_{max}$ and $k$ varied within their respective 95% confidence intervals. Since laboratory measurements of $P_G$ were conducted at a constant temperature of 20 °C and EC measurements at the air temperatures present at the field site, the temperature limitation of GPP was studied by

fitting to the GPP data a model similar to Eq. (3), but complemented with a Gaussian type temperature response (Maanavilja et al. 2011):

$$GPP = \frac{Pmax \; PPFD}{k + PPFD}(LAI + b) \exp\left(\frac{-0.5 \; (T_a - T_{opt})^2}{T_{tol}^2}\right) \qquad (4)$$

where $T_a$ is the air temperature, $T_{opt}$ is the temperature optimum of GPP and $T_{tol}$ is the temperature tolerance of GPP (deviation from the optimum at which GPP is 60% of the maximum). Using Eq. (4), GPP at 20°C and at ambient PAR and LAI was simulated for the study site over the growing season.

**3 Results**

**3.1 Cumulative growing season gross photosynthesis**

Fitting the temperature response curve of $R_{eco}$, Eq. (2) into the nighttime eddy covariance data yielded a reference respiration ($R_{ref}$) of 123 mg ($CO_2$) m$^{-2}$ h$^{-1}$ and $Q_{10}$ of 3.5. In the GPP model (Eq. (3)) fit, *Pmax* was 1721.8 mg $CO_2$ m$^{-2}$ h$^{-1}$, $k$ was 128.3 µmol m$^{-2}$ s$^{-1}$ and $b$ was 0.08. After gap-filling the GPP data (Eq. (3)), the resulting cumulative

growing season GPP estimate was 243 g C m$^{-2}$ (95 % confidence interval; 220-265 g C m$^{-2}$). In the GPP model fit complemented with temperature response (Eq. (4)), *Pmax* was 1852 mg ($CO_2$) m$^{-2}$ h$^{-1}$, $k$ was 170.3 µmol m$^{-2}$ s$^{-1}$, $b$ was 0.1, $T_{opt}$ was 22.6 °C and $T_{tol}$ was 20.9 °C.

Cumulative growing season $P_G$ upscaled to the ecosystem level using the separate light response curves for species and months (Eq. (1)) was 230 g C m$^{-2}$ (Julian days 121–273). Daily $P_G$ estimates were higher than GPP values from the EC tower in spring and lower in the middle of the summer (Fig. 1a). The GPP simulated at 20 °C, the same temperature as during the laboratory measurements, was similar than upscaled $P_G$ in spring but closer to the measured GPP in the middle of the summer (Fig. 1a). In the autumn, all of the three estimates showed rather similar levels (Fig. 1a).

*Sphagna* at the study site had higher cumulative growing season $P_G$ value (138 g C m$^{-2}$) than vascular plants (92 g C m$^{-2}$). *Sphagna* had higher daily $P_G$ than vascular plants in spring and autumn, but were almost at the same level in the middle of the summer (Fig. 2). A small increase in *Sphagnum* photosynthesis was observed during May (Fig. 2 and 3b) due to increment of daily PPFD towards midsummer. Otherwise, *Sphagnum* $P_G$ decreased steadily over the growing season (Fig. 2). Seasonal changes in vascular $P_G$ showed similar patterns than vascular LAI development, although the maximum $P_G$ was reached slightly earlier in the season than maximum LAI (Fig. 1a, Ic and 2).

The three vascular plant species having the highest $P_G$ in the ecosystem were *C. vulgaris*, *R. alba* and *A. polifolia*. *A. polifolia* was the most productive species in May and September, *R. alba* in June and July and *C. vulgaris* in August (Fig. 3a and 4a). With 13% cover altogether (Table 1), they formed 22% of the seasonal ecosystem $P_G$ and 56% of the vascular plant $P_G$ (Fig. 4). The three *Sphagnum* species with highest $P_G$ at the ecosystem level were *S. papillosum*, *S. fuscum* and *S. rubellum* (Fig. 3b and 4b). As with all of the *Sphagnum* species, their $P_G$ per ground area decreased steadily over the growing season (Fig. 2 and 3b). With 42% cover altogether (Table 1), they formed 40% of the seasonal ecosystem $P_G$, 67% of the $P_G$ of *Sphagnum* mosses (Fig. 4).

**3.2 Cumulative growing season net photosynthesis**

The aboveground vegetation of the study site was a carbon sink of 77 g C m$^{-2}$ over the growing season as estimated by $P_N$ value upscaled to ecosystem level using the species- and month-wise light response curves. $P_N$ results for *Sphagna* and vascular plants were reversed in comparison to $P_G$ estimates; $P_N$ of *Sphagna* was 20 g C m$^{-2}$ and vascular $P_N$ was 57 g C m$^{-2}$.

The same vascular plant species had the highest growing season $P_N$ and $P_G$; *R. alba, C. vulgaris* and *A. polifolia* had the highest $P_N$ estimates of 15.1, 9.1 and 8.4 g C m$^{-2}$, respectively (Table 1). These three species made up 57% of the total vascular $P_N$ and 42% of the whole ecosystem-level $P_N$.

*S. fuscum*, *S. papillosum*, and *S. majus* had the highest seasonal $P_N$ of *Sphagnum* species 7.4, 6.8 and 2.8 g C m$^{-2}$, respectively (Table 1). The $P_N$ of these three species was 85% of the total *Sphagnum* $P_N$ and 22% of the seasonal ecosystem $P_N$. Although having one of the highest coverage and $P_G$, *S. rubellum* was not among the three most productive species in terms of $P_N$.

**4 Discussion**

**4.1 Comparison of upscaled gross photosynthesis values with eddy covariance gross primary production estimates**

Accounting for the differences in photosynthetic parameters between species and between phases of the growing season appeared to accurately estimate ecosystem $P_G$ when upscaling species level measurements. *Sphagnum* mosses especially showed a large seasonal variation in their photosynthetic light response, which could be accounted for in this upscaling approach. The similarity of the $P_G$ estimates calculated with species-wise and monthly light response curves and GPP estimates derived from EC measurements (Fig. 1a), adds credibility to the methods used and indicates that the photosynthetic parameters measured under laboratory conditions are comparable with field measurements. Both methods carry their error sources. Annual $CO_2$ flux balances from EC measurements are prone to significant systematic bias, sometimes in excess of 30%, but usually between 10–30% of the cumulative flux (e.g. Baldocchi, 2003; Rannik et al., 2006). The underestimation of the EC fluxes implicit in the unclosed energy balance (70% for Siikaneva-1, unpublished data) might be partly compensated by the Kok effect that might be more significant than thought before, as indicated by Wehr et al. (2016). Our $P_G$ estimates include errors related to the LAI development measurements, visual species cover estimation, the conversion from cover to LAI, and the laboratory measurements of photosynthetic parameters. Although the shading of the moss layer by vascular plants may figure as a potential error source of $P_G$, upscaled with PPFD measured above the vegetation, it is not likely to be caused by the sparse cover of vascular plants at the site (Supplementary information, Fig. S3) with low seasonal maximum LAI (Fig. 1c). By taking into account the variation between samples in the 87 $P_N$ models (Eq. (1)) we aimed at more accurate estimation of the light response parameters. Nevertheless, our ecosystem-level $P_G$ estimate may contain bias caused by not accounting for the random effects of the 87 models in the upscaling procedure. The cumulative growing season $P_G$ of 230 g C m$^{-2}$ is very similar to the 205 g C m$^{-2}$ obtained by Alm et al., (1999) at an ombrotrophic bog site under similar climatic conditions and comparable water levels, but where the exceptionally dry conditions during the measured season reduced photosynthetic capacity of many *Sphagnum* species. Our growing season $P_G$ was considerably lower than the 500 g C m$^{-2}$ obtained by Moore et al., (2002) and Roulet et al., (2007) at a temperate ombrotrophic bog with much lower water table levels. While our value only covers the period from May until September, it falls just below the large range of annual GPP values (250 to 900 g C m$^{-2}$) measured with the EC method from seven northern peatland sites (Lund et al., 2010).

The shapes of $P_G$ and GPP development differed over the growing season, especially at the beginning of the summer, which is largely due to the constant temperature of 20 °C in our laboratory measurements (Fig. 1a). The constant temperature allowed us to investigate how the changes in species-specific photosynthetic parameters were affected by the seasonal changes in moisture conditions in the field. Since measuring of species-specific temperature responses of $P_G$ was unachievable due to the large number of species, we instead chose to model the temperature dependence of EC-derived GPP (Eq. (4), Fig. 1a). Our upscaled $P_G$ values were higher than GPP in May when vascular plants had a high capacity to use low light levels (low k value) and *Sphagna* had high Pmax (Fig. 1a) (Supplementary information, Table S1), but this was the case when temperatures in the field remained mostly below 20 °C and limited the measured GPP (Fig. 1b). The temperature limitation of measured GPP is demonstrated by the lower spring-time measured GPP in comparison with GPP simulated at 20 °C (Eq, (4), Fig. 1a). Both measured GPP and GPP simulated at 20 °C show higher levels than $P_G$ in July and August (Fig. 1a), for which the reason remains partly unclear. Because the difference between GPP and $P_G$ lasted for two months, the lack of vascular plant $P_G$ measurements in July can only partly explain this midsummer deviation between the two methods. In September, when *Sphagnum* Pmax values and k values of both vascular plants and *Sphagna* were at

their lowest, $P_G$, measured GPP and GPP at 20 °C were all similar despite the difference between air and laboratory temperatures (Fig. 1a). According to our results peatland photosynthesis is temperature limited, especially in spring; *Sphagna* had a high photosynthetic potential due to favourable moisture conditions at that time (Fig. 2), but the low field temperatures limited ecosystem-level GPP (Fig. 1a and b). Temperature limitation of spring-time photosynthesis is well known for boreal forests (Tanja et al., 2003; Ensminger et al., 2004), as well as for bog *Sphagna* (Moore et al. 2006). Mean annual temperature together with PPFD during the growing season are the most important factors explaining *Sphagnum* productivity at the global scale (Gunnarsson, 2005; Loisel et al., 2012), and the temperature optimum of *Sphagnum* photosynthesis is known to change over the growing season (Gaberščik and Martinčič, 1987). However, the temperature dependence and acclimatization of species-level photosynthesis in peatlands has been studied only with a few *Sphagnum* species (Gaberščik and Martinčič, 1987; Robroek et al., 2007).

**4.2 The contribution of plant species to ecosystem-level gross photosynthesis**

Among both *Sphagna* and vascular plants, the species with the highest seasonal upscaled $P_G$ (g C per m$^{-2}$ of ground area) – and hence the most productive species at the ecosystem scale – were also the ones with the highest areal cover. No inter-species differences in photosynthetic properties, either within vascular plants or *Sphagna*, could change this order. At the ecosystem scale, *Sphagna* covering on average 63% of the ground area had higher upscaled daily $P_G$ values for the whole summer than vascular plants covering only 24% despite the lower Pmax values of *Sphagna* (Supplementary information, Table S1). In ombrotrophic bogs, *Sphagna* are known to be the first group to start photosynthesizing in early spring (Moore et al., 2006), which was also evident at our site (Fig. 2). Combination of low Pmax values in July and September and high respiration rates in August and September (Supplementary information, Table S2) resulted in an almost linear decrease in ecosystem-scale *Sphagnum* $P_G$ over the growing season (Fig. 2). The seasonally decreasing *Sphagnum* $P_G$ is likely to reflect the change in the moisture conditions. Water table depth, which, together with precipitation, is known to be the most important moderator of *Sphagnum* photosynthesis (Hayward and Clymo 1983; Backéus 1988; Lindholm 1990; Nijp et al. 2014), decreased at the study site over the growing season (Fig. 1d).

Despite low Pmax values, *R. alba* was among the three vascular plant species with highest $P_G$ at the ecosystem scale due to its high cover at the site (Table 1). It also had a very sharp but short-lived LAI and $P_G$ peak at the end of June (Fig. 3a), which was largely the reason for the peak in vascular plant $P_G$ (Fig. 2), occurring slightly earlier in the season than maximum vascular LAI (Fig. 1c). Evergreen shrubs have been observed to be the second group to start photosynthesizing after *Sphagna* in spring (Moore et al., 2006). Similarly, the vascular plants with highest upscaled $P_G$ in May were the evergreen shrubs *A. polifolia* and *C. vulgaris* (Fig. 3a). The contributions of different species to total *Sphagnum* $P_G$ did not differ over the growing season (Fig. 4). Based on these observations, phenology and areal cover, rather than differences in photosynthetic parameters among species seems to be the key factor in determining the species with highest $P_G$ of a bog ecosystem.

**4.3 Ecosystem-level net photosynthesis**

The variation in photosynthetic properties changed the roles of the plant species in seasonal ecosystem-level carbon sink. Although *Sphagna* had more than twice the cover of vascular plants, seasonal $P_N$ was much lower than vascular plants (Table 1). The seasonal $P_N$ of *Sphagnum* species was not in relation with their areal cover; for

example, the species with highest cover, *S. rubellum*, had a small seasonal $P_N$ (Table 1). *S. rubellum* has earlier found to have lower light saturated photosynthesis and higher respiration than most of the other *Sphagnum* species (Supplementary information, Table S1, Korrensalo et al., 2016). The differences in photosynthetic parameters of *Sphagnum* species seem to become much more visible in ecosystem-level $P_N$ than of vascular plant species, since the leaf area of *Sphagna* stays similar over the growing season. The vascular plants most important for the ecosystem-level $P_N$ were the same as the species with greatest cover. However, *T. cespitosum* with only 1% of areal cover made up 12% of the seasonal vascular $P_N$.

Our results indicate that in addition to areal cover of the species, differences in photosynthetic parameters between species shape the ecosystem-level carbon sink of a bog. Species with low areal cover may be important for the ecosystem carbon sink because of their high photosynthetic efficiency. However, the $P_N$ results have to be interpreted with care, since they contain the R parameter estimated based on respiration measurements done at 20 °C, which is higher than field temperature for most of the growing season. This general overestimation of respiration may be the reason behind slightly negative seasonal $P_N$ of *S. balticum* (Table 1).

### 4.4 The role of functional diversity for peatland carbon sink

According to the insurance hypothesis, species diversity both enhances productivity and decreases the temporal variance of productivity of a plant community (Yachi and Loreau, 1999). This hypothesis has gained support from testing in several ecosystem types, especially in grasslands (Hector et al., 2010; Cardinale et al., 2011; Morin et al., 2014). In addition to species and genotype diversity (Hughes et al., 2008), the functional diversity, i.e. the presence of species and plant functional types with different physiology, morphology, resource requirements, seasonal growth patterns and life history may increase the productivity of an ecosystem (Tilman et al., 1997; Cadotte et al., 2008). Although this study did not directly test the insurance hypothesis, our results also indicate that functional diversity, especially in regard of differences in phenology and seasonal changes in photosynthetic parameters, decreased the temporal variation of ecosystem-level $P_G$ and could therefore decrease the variation of the ecosystem C sink. Vascular plant species of different phenology had the highest ecosystem-level photosynthesis at distinct phases of the growing season (Fig. 3a). In addition, *Sphagna* and evergreen shrubs formed two stable baselines of ecosystem $P_G$, which was further increased by the mid-summer $P_G$ peak of the sedge *R. alba* (Fig. 3 a and b). Especially ecosystem-level *Sphagnum* $P_G$ was modified by the seasonal decrease in photosynthetic potential (Supplementary information, Table S1). This suggests that the growing season $P_G$ of our study site is not only more stable, but it is also larger than it would be with a more functionally homogenous assemblage of species. Several studies have suggested that the C sink function of bogs is more stable over the growing season than that of fens, which have more homogenous and sedge dominated vegetation (Bubier et al., 1998; Leppälä et al., 2008). Hence, our results should be compared with the patterns of photosynthetic productivity of a peatland site with a more homogenous plant assemblage.

Based on small-scale experimental studies, bog species and growth forms are known to vary in terms of their contribution to ecosystem productivity and to differ in their responses to manipulations of environmental conditions (Weltzin et al., 2000; Ward et al., 2009; Kuiper et al., 2014). In this study, the laboratory measurements of species photosynthetic parameters were for the first time upscaled to ecosystem level over a whole growing season and verified by the comparison with EC measurements. Boreal bog species were found to differ in the

timing of their maximum $P_G$ over a growing season (Fig. 3a and b). Diversity in species responses to environmental factors is hypothesized to make a plant community more resilient towards changing conditions (Yachi and Loreau, 1999; Gunderson, 2000). In addition to species diversity, plant community diversity within an ecosystem has been shown to increase ecosystem stability during a severe drought in grasslands (Frank and McNaughton, 1991). In boreal bogs, *Sphagnum* mosses create microtopographic variations that – according to model simulations – increase resilience towards environmental perturbations both through the diversity of growth forms it supports and by variation in physical properties between microforms (Turetsky et al., 2012). To find out about the effect of bog spatial heterogeneity on ecosystem resilience, studies extending over several growing seasons are needed. As demonstrated in an arctic sedge fen, the impact of extreme weather conditions on ecosystem C sink may occur with a lag of one growing season (Zona et al., 2014). Our study provides tools to empirically study the role of species and community diversity at the ecosystem scale. The combination of laboratory measurements of photosynthetic parameters, phenological monitoring and EC measurements opens up the possibility of long-term and experimental ecosystem-level studies on the effect of functional diversity on the peatland ecosystem carbon sink. The long-term measurements would permit the inclusion of a wider range of environmental conditions. In particular, the EC method would allow for a comparison of the effect of diversity at sites with different plant assemblages.

**4.5 Conclusions**

The areal cover of the species determined the species with the highest gross photosynthesis while phenology in leaf area and photosynthetic activity drove the variation in ecosystem-level gross photosynthesis. In spring, potential ecosystem-level gross photosynthesis was much higher than measured gross primary production, which appeared to be due to temperature limitation of photosynthesis. Ecosystem-level net photosynthesis was more of a combination of the differences in (i) photosynthetic parameters, which were important in *Sphagna*, (ii) phenology, which largely defined vascular productivity, and (iii) areal coverage, which acted in both vascular plants and *Sphagna*.

The different growth strategies of the plant species present at our study site appeared to increase the ecosystem-level photosynthesis and decrease it's variation within a growing season. We are looking forward for the future studies finding out, if the diversity of growth forms has the same stabilizing effect on the interannual variation of ecosystem-level photosynthesis.

**Data availability**

The data associated with the manuscript will be published in PANGAEA repository. Upon request, the data can also be obtained from the corresponding author.

**Author contribution**

EST formulated the idea. AK, TH and EST designed the measurements, which were done by AK and TH. AK, TH and EST were responsible for the primary photosynthesis data processing. Eddy covariance data collection and analysis was done by PA, JR, TV and IM. The mixed-effects models were developed by LM and AK. AK, PA and EST wrote the manuscript, which was commented by all the other authors.

**Acknowledgements**

The work presented here is supported by the Faculty of Science and Forestry in the University of Eastern Finland, the Finnish Cultural Foundation, the Academy of Finland (287039, 118780, 1284701, 1282842), ICOS (271878), ICOS-Finland (281255) and the Nordic Centre of Excellence – DEFROST. We would also like to thank the staff

at Hyytiälä Forest Research Station and Salli Uljas, Janne Sormunen, María Gutierrez, Laura Kettunen and Eva-Stina Kerner for their help with the measurements and Nicola Kokkonen for revising the English language of the manuscript.

**Conflict of Interest**

The authors declare that they have no conflict of interest.

**References**Alm, J., Schulman, L., Walden, J., Nykänen, H., Martikainen, P. J. and Silvola, J.: Carbon balance of a boreal bog during a year with an exceptionally dry summer, Ecology, 80(1), 161–174, doi:10.1890/0012-9658(1999)080[0161:CBOABB]2.0.CO;2, 1999.

Aubinet, M., Vesala, T. and Papale, D.: Eddy Covariance: A Practical Guide to Measurement and Data Analysis, Springer Netherlands, 2012.

Backéus, I.: Weather variables as predictors of *Sphagnum* growth on a bog, Holarctic Ecology, 11(2), 146–50. 1988.

Baldocchi, D. D.: Assessing the eddy covariance technique for evaluating carbon dioxide exchange rates of ecosystems: past, present and future, Global Change Biology, 9(4), 479–492, 2003.

Breeuwer, A., Robroek, B. J. M., Limpens, J., Heijmans, M. M. P. D., Schouten, M. G. C. and Berendse, F.: Decreased summer water table depth affects peatland vegetation, Basic and Applied Ecology, 10(4), 330–339, doi:10.1016/j.baae.2008.05.005, 2009.

Bubier, J., Crill, P., Moore, T., Savage, K. and Varner, R.: Seasonal patterns and controls on net ecosystem $CO_2$ exchange in a boreal peatland complex, Global Biogeochemical Cycles, 12, 703–714, doi: 10.1029/98GB02426,

1998.

Cadotte, M. W., Cardinale, B. J. and Oakley, T. H.: Evolutionary history and the effect of biodiversity on plant productivity, Proceedings of the National Academy of Sciences, 105(44), 17012–17017, 2008.

Cardinale, B. J., Matulich, K. L., Hooper, D. U., Byrnes, J. E., Duffy, E., Gamfeldt, L., Balvanera, P., O'Connor,

30    M. I. and Gonzalez, A.: The functional role of producer diversity in ecosystems, American Journal of Botany, 98(3), 572–592, doi:10.3732/ajb.1000364, 2011.

Chapin, F.S., Matson, P.A., Vitousek, P.M., and Chapin, M.C. : Principles of terrestrial ecosystem ecology. 2nd

ed. Springer, New York, N.Y., 1991.

Ensminger, I., Sveshnikov, D., Campbell, D. A., Funk, C., Jansson, S., Lloyd, J., Shibistova, O. and Ãquist, G.: Intermittent low temperatures constrain spring recovery of photosynthesis in boreal Scots pine forests, Global Change Biology, 10(6), 995–1008, doi:10.1111/j.1365-2486.2004.00781.x, 2004.

Frank, D. A. and McNaughton, S. J.: Stability Increases with Diversity in Plant Communities: Empirical Evidence from the 1988 Yellowstone Drought, Oikos, 62(3), 360–362, doi:10.2307/3545501, 1991.

Frolking, S. E., Bubier, J. L., Moore, T. R., Ball, T., Bellisario, L. M., Bhardwaj, A., Carroll, P., Crill, P. M., Lafleur, P. M., McCaughey, J. H., Roulet, N. T., Suyker, A. E., Verma, S. B., Waddington, J. M. and Whiting, G. J.: Relationship between ecosystem productivity and photosynthetically active radiation for northern peatlands, Global Biogeochemical Cycles, 12, 115–126, doi: 10.1029/97GB03367, 1998

Gaberščik, A. and Martinčič, A.: Seasonal dynamics of net photosynthesis and productivity of Sphagnum papillosum, Lindbergia, 105–110, 1987.

Gorham, E.: Northern Peatlands: Role in the Carbon Cycle and Probable Responses to Climatic Warming, Ecological Applications, 1(2), 182, doi:10.2307/1941811, 1991.

Gunderson, L. H.: Ecological resilience–in theory and application, Annual review of ecology and systematics, 425–439, 2000.

Gunnarsson, U.: Global patterns of *Sphagnum* productivity, Journal of Bryology, 27(3), 269–279, doi:10.1179/174328205X70029, 2005.

Hájek, T., Tuittila, E.-S., Ilomets, M. and Laiho, R.: Light responses of mire mosses - a key to survival after water-level drawdown?, Oikos, 118(2), 240–250, doi:10.1111/j.1600-0706.2008.16528.x, 2009.

Hayward, P.M., and Clymo, R.S.: The growth of *Sphagnum*: experiments on, and simulation of, some effects of light flux and water-table depth, Journal of Ecology, 71(3), 845–863, 1983.

Hämet-Ahti, L., Suominen, J., Ulvinen, T. and Uotila, P. (eds.): Retkeilykasvio (Field Flora of Finland), Ed. 4. Finnish Museum of Natural History, Helsinki, 1998.

Hector, A., Hautier, Y., Saner, P., Wacker, L., Bagchi, R., Joshi, J., Scherer-Lorenzen, M., Spehn, E. M., Bazeley-White, E., Weilenmann, M. and others: General stabilizing effects of plant diversity on grassland productivity through population asynchrony and overyielding, Ecology, 91(8), 2213–2220, 2010.

Hughes, A. R., Inouye, B. D., Johnson, M. T. J., Underwood, N. and Vellend, M.: Ecological consequences of genetic diversity: Ecological effects of genetic diversity, Ecology Letters, 11(6), 609–623, doi:10.1111/j.1461-0248.2008.01179.x, 2008.

Kormann, R. and Meixner, F. X.: An analytical footprint model for non-neutral stratification, Boundary-Layer Meteorology, 99(2), 207–224, 2001.

Korrensalo, A., Hájek, T., Vesala, T., Mehtätalo, L. and Tuittila, E-S.: Variation in photosynthetic properties among bog plants, Botany, doi:10.1139/cjb-2016-0117, (in press).

Kuiper, J. J., Mooij, W. M., Bragazza, L. and Robroek, B. J.: Plant functional types define magnitude of drought

response in peatland CO2 exchange, Ecology, 95(1), 123–131, 2014.

Laine, A. M., Ehonen, S., Juurola, E., Mehtätalo, L. and Tuittila, E.-S.: Performance of late succession species along a chronosequence: Environment does not exclude *Sphagnum fuscum* from the early stages of mire development, edited by M. Zobel, Journal of Vegetation Science, 26(2), 291–301, doi:10.1111/jvs.12231, 2015.

Laine, J.: The intricate beauty of "Sphagnum" mosses: a Finnish guide to identification, University of Helsinki

Department of Forest Ecology, 2009.

Larcher, W.: Physiological Plant Ecology: Ecophysiology and Stress Physiology of Functional Groups, Springer, 2003.

Leppälä, M., Kukko-Oja, K., Laine, J. and Tuittila, E.-S.: Seasonal dynamics of $CO_2$ exchange during primary succession of boreal mires as controlled by phenology of plants, Ecoscience, 15(4), 460–471, doi:10.2980/15-4-

3142, 2008.

Lindholm, T.: Growth dynamics of the peat moss *Sphagnum fuscum* on hummocks on a raised bog in southern Finland, Annales Botanici Fennici, 27, 67-78, 1990.

Loisel, J., Gallego-Sala, A. V. and Yu, Z.: Global-scale pattern of peatland *Sphagnum* growth driven by

photosynthetically active radiation and growing season length, Biogeosciences, 9(7), 2737–2746, doi:10.5194/bg-9-2737-2012, 2012.

Lund, M., Lafleur, P. M., Roulet, N. T., Lindroth, A., Christensen, T. R., Aurela, M., Chojnicki, B. H., Flanagan, L. B., Humphreys, E. R., Laurila, T., Oechel, W. C., Olejnik, J., Rinne, J., Schubert, P. and Nilsson, M. B.: Variability in exchange of CO2 across 12 northern peatland and tundra sites: exchange of $CO_2$ in wetlands,

Global Change Biology, 16(9), 2436-2448, doi:10.1111/j.1365-2486.2009.02104.x, 2009.

Lund, M., Christensen, T. R., Lindroth, A. and Schubert, P.: Effects of drought conditions on the carbon dioxide dynamics in a temperate peatland, Environmental Research Letters, 7(4), 45704, doi:10.1088/1748-9326/7/4/045704, 2012.

Maanavilja, L., Riutta, T., Aurela, M., Pulkkinen, M., Laurila, T. and Tuittila, E.-S.: Spatial variation in CO2

exchange at a northern aapa mire, Biogeochemistry, 104(1–3), 325–345, doi:10.1007/s10533-010-9505-7, 2011.

Mammarella, I., Peltola, O., Nordbo, A., Järvi, L., and Rannik, Ü.: Quantifying the uncertainty of eddy covariance fluxes due to the use of different software packages and combinations of processing steps in two contrasting ecosystems, Atmospheric Measurement Techniques, 9, 4915-4933, doi:10.5194/amt-9-4915-2016, 2016.

de Miguel, S., Mehtätalo, L., Shater, Z., Kraid, B. and Pukkala, T.: Evaluating marginal and conditional predictions of taper models in the absence of calibration data, Canadian Journal of Forest research, 42(7), 1383-1394, doi: 10.1139/x2012-090, 2012.

Moore, T. R., Bubier, J. L., Frolking, S. E., Lafleur, P. M. and Roulet, N. T.: Plant biomass and production and $CO_2$ exchange in an ombrotrophic bog, Journal of Ecology, 90(1), 25–36, doi:10.1046/j.0022-0477.2001.00633.x, 2002.

Moore, T. R., Lafleur, P. M., Poon, D. M. I., Heumann, B. W., Seaquist, J. W. and Roulet, N. T.: Spring photosynthesis in a cool temperate bog, Global Change Biology, 12(12), 2323–2335, doi:10.1111/j.1365-2486.2006.01247.x, 2006.

Morin, X., Fahse, L., de Mazancourt, C., Scherer-Lorenzen, M. and Bugmann, H.: Temporal stability in forest productivity increases with tree diversity due to asynchrony in species dynamics, edited by M. Rejmanek, Ecology Letters, 17(12), 1526–1535, doi:10.1111/ele.12357, 2014.

Nijp, J. J., Limpens, J., Metselaar, K., van der Zee, S. E. A. T. M., Berendse, F. and Robroek, B. J. M.: Can frequent precipitation moderate the impact of drought on peatmoss carbon uptake in northern peatlands?, New Phytologist, 203(1), 70–80, doi:10.1111/nph.12792, 2014.

Pinheiro, J. and Bates, D.: Mixed-Effects Models in S and S-PLUS, Springer New York, 2000.

Rannik, Ü., Kolari, P., Vesala, T. and Hari, P.: Uncertainties in measurement and modelling of net ecosystem exchange of a forest, Agricultural and Forest Meteorology, 138(1–4), 244–257, doi:10.1016/j.agrformet.2006.05.007, 2006.

Robroek, B. J. M., Limpens, J., Breeuwer, A. and Schouten, M. G. C.: Effects of water level and temperature on performance of four Sphagnum mosses, Plant Ecology, 190(1), 97–107, doi:10.1007/s11258-006-9193-5, 2007.

Roulet, N. T., Lafleur, P. M., Richard, P. J. H., Moore, T. R., Humphreys, E. R. and Bubier, J.: Contemporary carbon balance and late Holocene carbon accumulation in a northern peatland, Global Change Biology, 13(2), 397–411, doi:10.1111/j.1365-2486.2006.01292.x, 2007.

Rydin, H. and Jeglum, J. K.: The Biology of Peatlands, OUP Oxford, 2013.

Small, E.: Photosynthetic rates in relation to nitrogen recycling as an adaptation to nutrient deficiency in peat bog plants, Canadian Journal of Botany, 50(11), 2227–2233, 1972.

Tanja, S., Berninger, F., Vesala, T., Markkanen, T., Hari, P., Mäkelä, A., Ilvesniemi, H., Hänninen, H., Nikinmaa, E., Huttula, T., Laurila, T., Aurela, M., Grelle, A., Lindroth, A., Arneth, A., Shibistova, O., Lloyd, J.: Air temperature triggers the recovery of evergreeen boreal forest photosynthesis in spring, Global Change Biology, 9, 1410-1426, 2003.

Tilman, D., Knops, J., Wedin, D., Reich, P., Ritchie, M. and Siemann, E.: The Influence of Functional Diversity and Composition on Ecosystem Processes, Science, 277(5330), 1300, doi:10.1126/science.277.5330.1300, 1997.

Turetsky, M. R., Bond-Lamberty, B., Euskirchen, E., Talbot, J., Frolking, S., McGuire, A. D. and Tuittila, E.-S.: The resilience and functional role of moss in boreal and arctic ecosystems: Tansley review, New Phytologist, 196(1), 49–67, doi:10.1111/j.1469-8137.2012.04254.x, 2012.

Waddington, J. M. and Roulet, N. T.: Carbon balance of a boreal patterned peatland, Global Change Biology, 5  6(1), 87–97, doi:10.1046/j.1365-2486.2000.00283.x, 2000.

Ward, S. E., Bardgett, R. D., McNamara, N. P. and Ostle, N. J.: Plant functional group identity influences short-term peatland ecosystem carbon flux: evidence from a plant removal experiment, Functional Ecology, 23(2), 454–462, doi:10.1111/j.1365-2435.2008.01521.x, 2009.

Wehr, R., Munger, J. W., McManus, J. B., Nelson, D. D., Zahniser, M. S., Davidson, E. A., 10  Wofsy, S. C. and Saleska, S. R.: Seasonality of temperate forest photosynthesis and daytime respiration, Nature, 534(7609), 680–683, 2016.

Weltzin, J. F., Pastor, J., Harth, C., Bridgham, S. D., Updegraff, K. and Chapin, C. T.: Response of bog and fen plant communities to warming and water-table manipulations, Ecology, 81(12), 3464–3478, doi:10.1890/0012-15  9658(2000)081[3464:ROBAFP]2.0.CO;2, 2000.

Wilson, D., Alm, J., Riutta, T., Laine, J., Byrne, K. A., Farrell, E. P. and Tuittila, E.-S.: A high resolution green area index for modelling the seasonal dynamics of CO2 exchange in peatland vascular plant communities, Plant Ecology, 190(1), 37–51, doi:10.1007/s11258-006-9189-1, 2007.

Yachi, S. and Loreau, M.: Biodiversity and ecosystem productivity in a fluctuating environment: the insurance 20  hypothesis, Proceedings of the National Academy of Sciences, 96(4), 1463–1468, 1999.

Yu, Z. C.: Northern peatland carbon stocks and dynamics: a review, Biogeosciences, 9(10), 4071–4085, doi:10.5194/bg-9-4071-2012, 2012.

Zona, D., Lipson, D. A., Richards, J. H., Phoenix, G. K., Liljedahl, A. K., Ueyama, M., Sturtevant, C. S. and Oechel, W. C.: Delayed responses of an Arctic ecosystem to an extreme summer: impacts on net ecosystem 25  exchange and vegetation functioning, Biogeosciences, 11(20), 5877–5888, doi:10.5194/bg-11-5877-2014, 2014.

**Tables**

**Table 1** Average projection cover of the most common plant species at the study site, maximum leaf area index (LAI) values, and cumulative seasonal gross and net photosynthesis ($P_G$, $P_N$) of the species measured in this study. The cover values are based on a vegetation inventory conducted in 2013 at the study site within the 30 m radius footprint of the eddy covariance tower (N=121). LAI values of vascular plants are seasonal maxima of each species calculated by converting the cover values into LAI using species-wise linear relationships (Supplementary information, Table S2). *Sphagnum* LAI is assumed to stay similar over the growing season and is simply the coverage expressed as LAI. *Sphagnum* and vascular species nomenclature according to Laine et al., (2009) and Hämet-Ahti (1998), respectively.

| Species | Cover mean ± S.E. (%) | LAI ($m^2$ $m^{-2}$) | Seasonal $P_G$ (g C $m^{-2}$) | Seasonal $P_N$ (g C $m^{-2}$) |
|---|---|---|---|---|
| Vascular total | 24.2 ± 1.9 | 0.29 | 92.2 | 57.3 |
| *Rhynchospora alba* | 6.9 ± 0.8 | 0.10 | 23.3 | 15.1 |
| *Andromeda polifolia* | 3.7 ± 0.4 | 0.03 | 14.4 | 8.4 |
| *Calluna vulgaris* | 2.8 ± 0.9 | 0.04 | 13.8 | 9.1 |
| *Rubus chamaemorus* | 2.5 ± 0.6 | 0.03 | 6.9 | 4.5 |
| *Eriophorum vaginatum* | 1.5 ± 0.3 | 0.02 | 6.0 | 3.8 |
| *Vaccinium oxycoccos* | 1.2 ± 0.2 | 0.01 | 5.3 | 3.2 |
| *Drosera rotundifolia* | 1.1 ± 0.1 | | | |
| *Empetrum nigrum* | 1.0 ± 0.5 | 0.01 | 2.3 | 1.2 |
| *Trichophorum cespitosum* | 1.0 ± 0.5 | 0.02 | 11.5 | 6.7 |
| *Drosera longifolia* | 0.8 ± 0.4 | | | |
| *Scheuchzeria palustris* | 0.8 ± 0.1 | 0.02 | 5.1 | 3.0 |
| *Betula nana* | 0.4 ± 0.2 | 0.004 | 1.1 | 0.5 |
| *Carex limosa* | 0.4 ± 0.1 | 0.005 | 2.4 | 1.8 |
| *Sphagnum* total | 63.8 ± 3.7 | 0.65 | 137.8 | 19.9 |
| *Sphagnum rubellum* | 18.3 ± 2.6 | 0.18 | 30.7 | 0.8 |
| *S. papillosum* | 12.9 ± 2.3 | 0.13 | 33.9 | 6.8 |
| *S. fuscum* | 11.0 ± 2.3 | 0.11 | 27.1 | 7.4 |
| *S. balticum* | 8.3 ± 1.5 | 0.08 | 15.7 | -0.5 |
| *S. cuspidatum* | 4.8 ± 1.3 | 0.05 | 13.4 | 1.7 |
| *S. majus* | 4.7 ± 1.2 | 0.05 | 12.7 | 2.8 |
| *S. angustifolium* | 1.3 ± 0.5 | 0.01 | 3.6 | 0.6 |
| *S. lindbergii* | 0.8 ± 0.8 | | | |
| *S. magellanicum* | 0.3 ± 0.1 | 0.003 | 0.7 | 0.1 |
| Other mosses and lichens | | | | |
| *Pleurozium schreberi* | 0.8 ± 0.5 | | | |
| *Mylia anomala* | 0.2 ± 0.1 | | | |
| *Cladina rangiferina* | 0.4 ± 0.2 | | | |

**Figures**

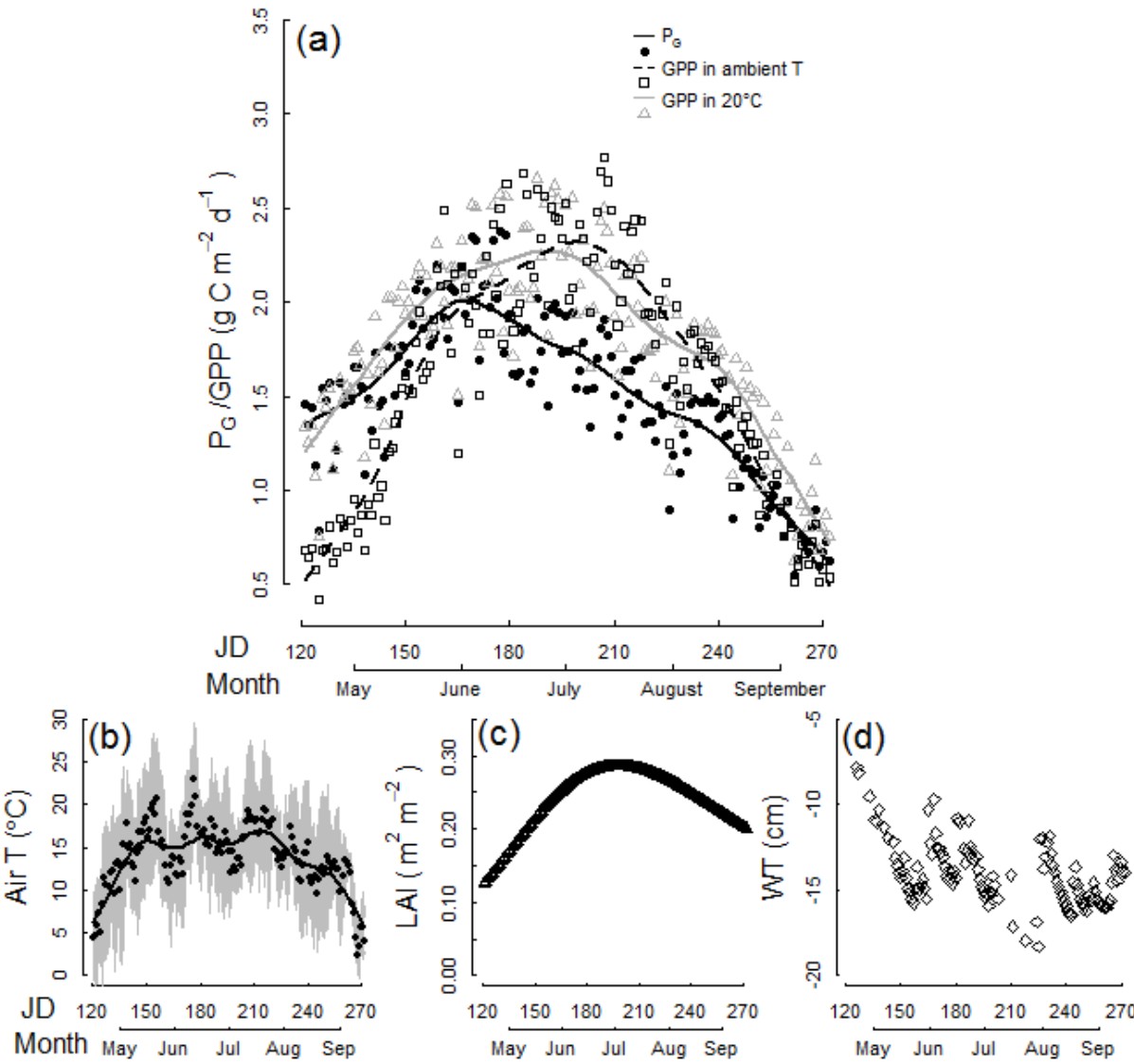

**Figure 1:** a) Comparison of total daily ecosystem-level gross photosynthesis estimate of all plants (P$_G$) derived from laboratory measurements with GPP estimates derived directly from the eddy covariance measurements and with GPP simulated at constant temperature of 20°C. The temperature in laboratory photosynthesis measurements was kept constant at 20 °C during the whole growing season. b) Daily mean air temperature (in black) and daily temperature variation (in grey fill) (Hyytiälä Forest Research Station 10 km from the study site, Finnish Meteorological Institute, 2016), c) sum of modeled vascular leaf area during the growing season 2013 and d) average daily lawn (i.e. intermediate) surface water table (WT) at the study site. Lines represent locally weighted scatterplot smoothing (Loess, smoothing parameter=0.25) curves.

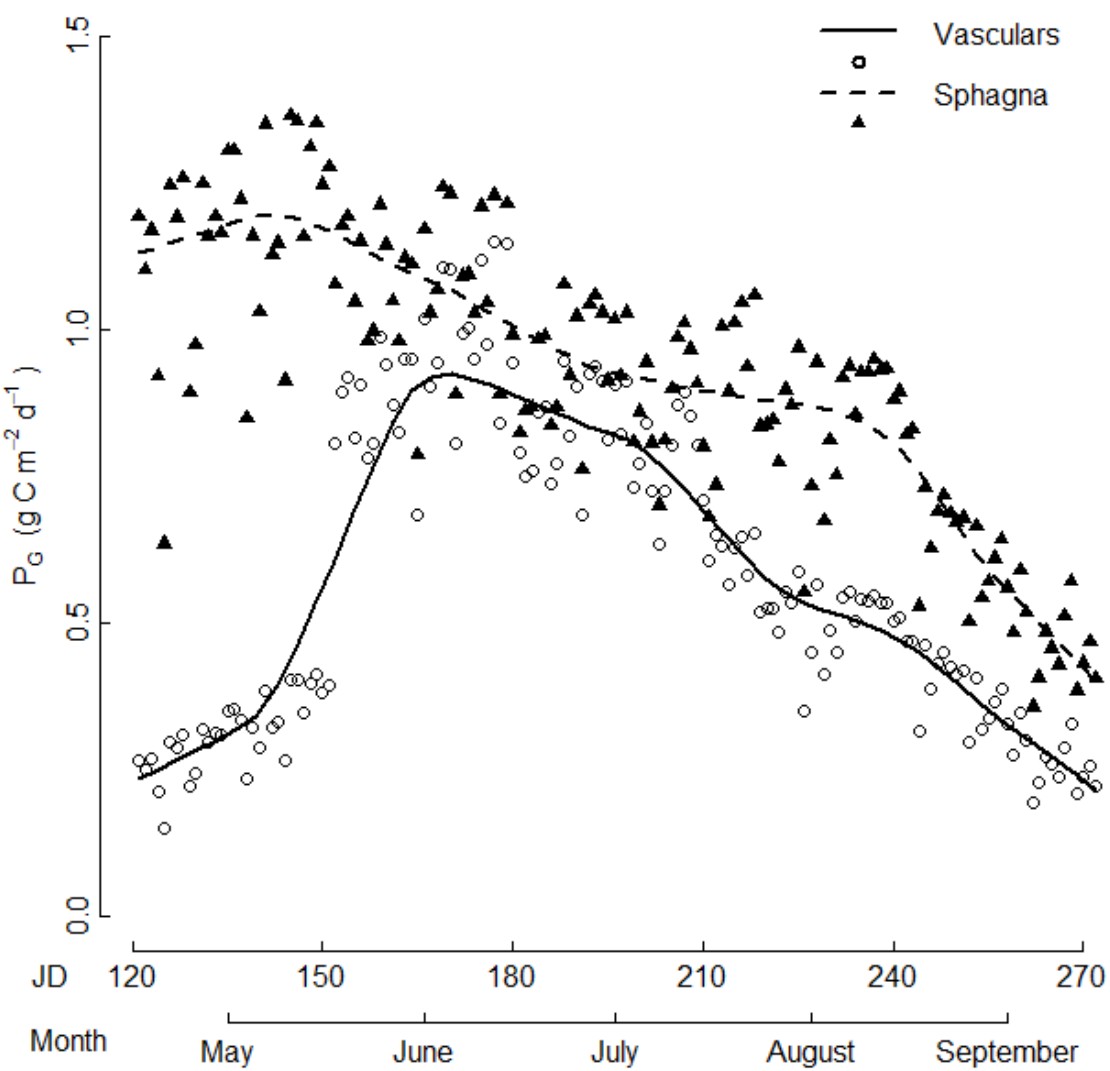

**Figure 2:** Daily gross photosynthesis estimates (g C m$^{-2}$ d$^{-1}$) of vascular plants and *Sphagna* upscaled to ecosystem-level using the species-specific, monthly light response curves derived from laboratory measurements. Lines represent Loess averaging (smoothing parameter=0.25)

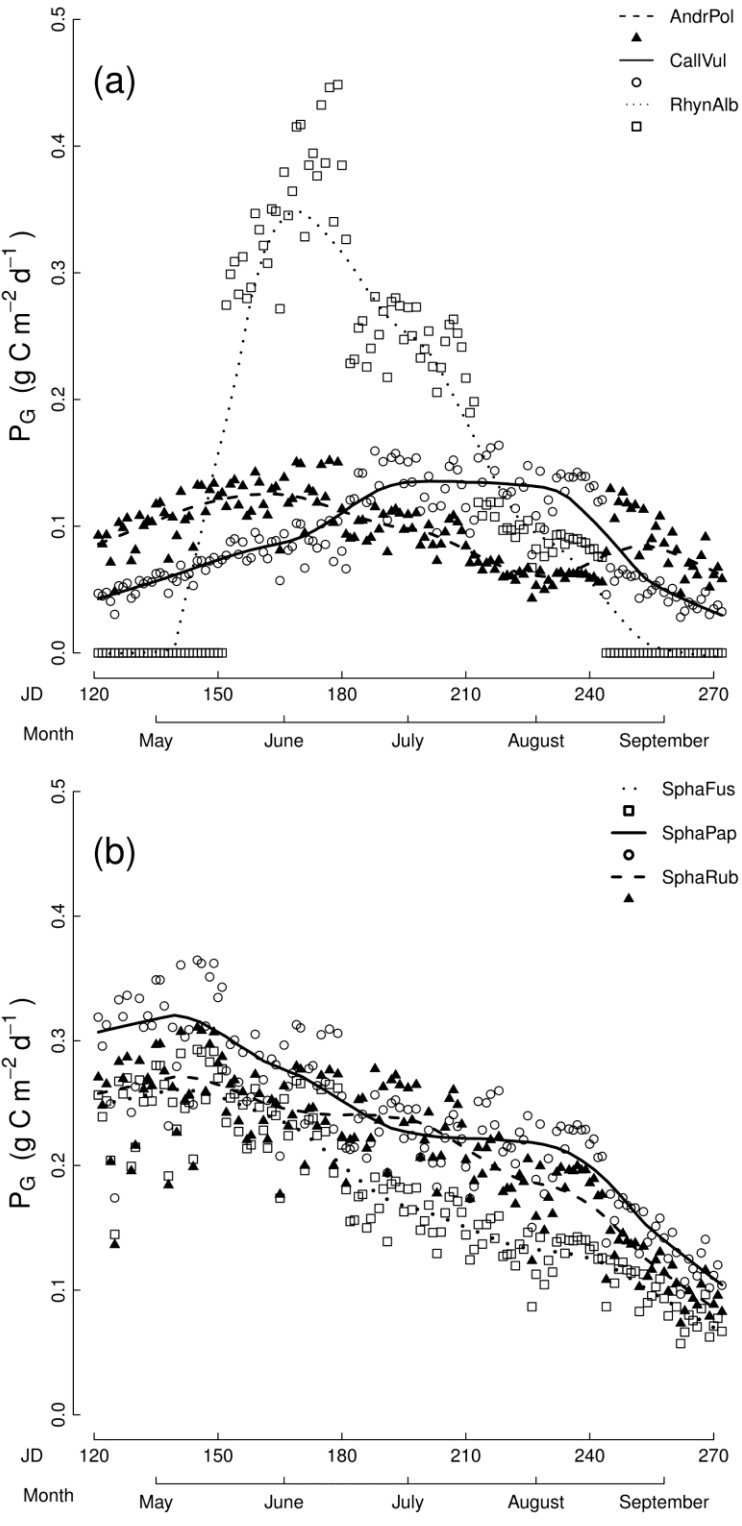

**Figure 3:** Daily gross photosynthesis estimates (g C m$^{-2}$ d$^{-1}$) of the three most productive a) vascular and b) *Sphagnum* species upscaled to ecosystem-level using the species-specific, monthly light response curves derived from laboratory measurements. Lines represent Loess averaging (smoothing parameter=0.25). The species cover within the study site (EC footprint) is given in Table 1. Abbreviations of the species' names are: AndrPol=*Andromeda polifolia*, CallVul=*Calluna vulgaris*, RhynAlb=*Rhynchospora alba*, SphaFusc=*Sphagnum fuscum*, SphaPapi=*Sphagnum papillosum*, SphaRube=*Sphagnum rubellum.*

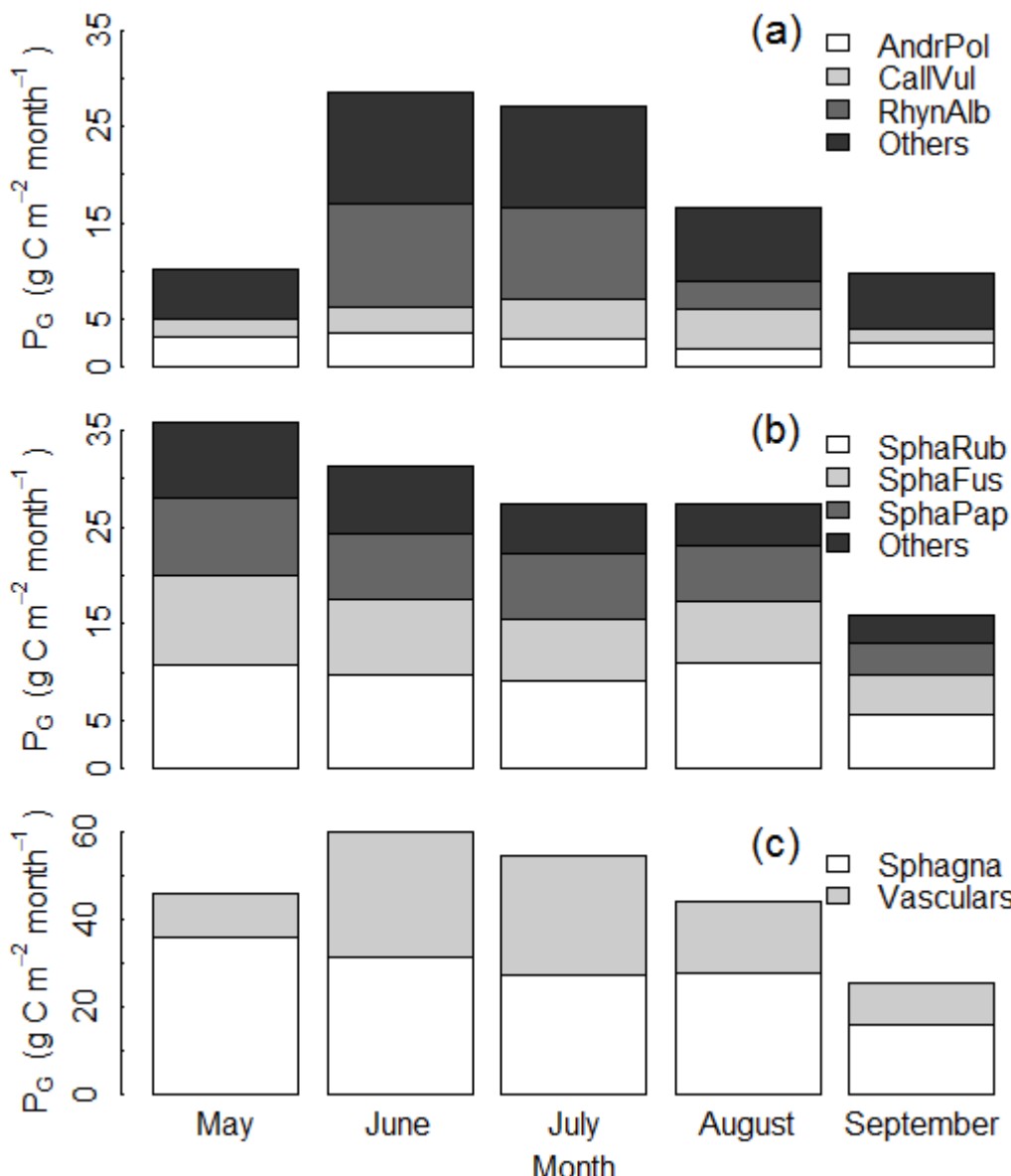

**Figure 4:** Species' proportions of monthly ecosystem-scale gross photosynthesis ($P_G$) of a) vascular plants, b) *Sphagnum* mosses and c) the contribution of those two groups to total monthly ecosystem scale $P_G$. Ecosystem-level $P_G$ was calculated using the species-specific, monthly light response curves derived from laboratory measurements. For abbreviations of the species' names see Figure 3.