# Peer review of "Species-specific temporal variation in photosynthesis as a moderator of peatland carbon sequestration"

_Biogeosciences, 2016_

## Referee Comment (RC1) · Anonymous Referee #1 · 18 Aug 2016

Korrensalo et al. presents one season of field measurements (eddy covariance), controlled laboratory experiments and modeled results of net and gross photosynthesis rates and/or gross primary production from a boreal bog in southern Finland and emphasizes the species specific contributions and the integration of plot to ecosystem scales. In particular, Korrensalo et al. differentiate between the vascular and bryophyte (moss) contributions, where the latter is oftentimes given a shadow-role in the literature of carbon and energy fluxes. Here, mosses are emphasized to play an important role in the overall wetland ecosystem-level flux. The two approaches in reaching the total system fluxes (eddy covariance and species-specific laboratory experiments) arrive at similar total seasonal fluxes. However, the figures suggest rather large seasonal differences (if I interpret them correctly). I would therefore appreciate increased attention to why that is. In fact, I think this difference is an interesting story (the story?) that emerged. Below are some thoughts that came to me as I reviewed.

Please define gross primary production (GPP), net (PN) and gross (PG) photosynthesis so reader who are not regularly working with these terms can follow your manuscript.

All figures: The graphs are presented with units, but there are no labels on the y-axises. Please include labels.

Figure 1a: Why the discrepancy between the "total" and "eddy covariance" in Figure 1? It is unclear from the figure caption, but I think the two curves represent the laboratory derived estimate (total) and the eddy covariance estimate (eddy covariance) of the same variable? So why is the Total > EC in early season and EC > Total in later season?

Figure 1d: What does "daily lawn surface water table" represent? I am confused by the word "lawn" (makes me think of a golf course). I suggest removing the smoothing curve and not include any line between dots unless the dots represents continuous daily measurements of water levels (there seems to be a larger data gap around Julian day 210).

Figure 1b: I suggest plotting mean daily air temperature and then present the min and max daily air temperature as a shaded fill behind the mean daily air temperature line.

Figure 1 (figure caption): Why keeping the laboratory temperature at 20°C during the entire growing season if the mean daily air temperature only reached 20°C during a few days? What is the implication of this approach on the analyses? Can this partly explain the offsets in Fig 1a? We see a large drop in water tables in the field site following Julian day 120. The laboratory measurements tried to keep the temperature and moisture contents constant throughout the season, while the field measurements of air

[Figure]

temperature and water table (ie moisture) present rather large variations. How does the limited moisture variability of the laboratory approach affect the overall conclusions stated by the authors? I am worried the authors may have over-stated their findings due to the complex relationships between water, air temperature and photosynthesis found in the field setting, especially considering the deviations in Figure 1a. In combination with Figure 2 (which I assume is based upon laboratory analyses, please clarify in figure text), it looks to me like the vascular plants may have been water-limited (too much water) in their photosynthesis in early season in the field (??)

Figure 3b. Why the decreasing response of the Sphagnum species throughout the study period? The total seasonal gross photosynthesis is similar between the two methods, but the distribution of those fluxes over the season is rather different between the two methods (laboratory versus eddy covariance). This observation is currently not discussed in the manuscript and I think this is the most interesting piece of the results.

I would like to see the text in the results section to address the seasonal variability that we see in the figures. The results section is currently focusing on the total seasonal values, while the figures show some rather interesting seasonal variations (in time and between methods).

Please refer to specific figures in the discussion.

Page 7, Line 33: The sentence is odd. Remove "when" perhaps?

The discussion refers to time by naming the month. I suggest all graphs use months instead of Julian day.

The discussion or literature does not address the impact on hydrology to the photosynthesis, which, especially for mosses, can have a major impact.

―――――――――――――――――

---

## Author Comment (AC1) · 22 Aug 2016

Korrensalo et al. presents one season of field measurements (eddy covariance), controlled laboratory experiments and modeled results of net and gross photosynthesis rates and/or gross primary production from a boreal bog in southern Finland and emphasizes the species specific contributions and the integration of plot to ecosystem scales. In particular, Korrensalo et al. differentiate between the vascular and bryophyte (moss) contributions, where the latter is oftentimes given a shadow-role in the literature of carbon and energy fluxes. Here, mosses are emphasized to play an important role in the overall wetland ecosystem-level flux. The two approaches in reaching the total system fluxes (eddy covariance and species-specific laboratory experiments) arrive at similar total seasonal fluxes. However, the figures suggest rather large seasonal differences (if I interpret them correctly). I would therefore appreciate increased attention to why that is. In fact, I think this difference is an interesting story (the story?) that emerged. Below are some thoughts that came to me as I reviewed.

> *We thank the reviewer for the time and effort used to our manuscript and for the thoughtful comments, which have been addressed one by one below. We agree with the reviewer that the difference between the two methods should be discussed more clearly than it has been done before. We hope the suggested clarifications below sufficiently bring up the reason behind these differences.*

> *However, we would be more than glad to hear through the discussion forum of the journal if the reviewer thinks we have not fully understood her/his ideas or do not answer the questions with sufficient accuracy.*

Please define gross primary production (GPP), net (PN) and gross (PG) photosynthesis so reader who are not regularly working with these terms can follow your manuscript.

> *Thank you for the comment. We will add the definitions of these terms (below) in the next version of the manuscript.*

> *"In the scale of individual plant leaves, net photosynthesis ($P_N$) is the $CO_2$ gain of the leaves, which is the leaf respiration (R) subtracted from gross photosynthesis ($P_G$). In the ecosystem scale, net ecosystem exchange (NEE) of $CO_2$ between the ecosystem and the atmosphere consists of ecosystem respiration ($R_{eco}$) subtracted from gross primary productivity (GPP). GPP is the rate by which $CO_2$ enters the ecosystem through $P_G$ of all the individual leaves together (Chapin et al. 2011)."*

> Chapin, F.S., Matson, P.A., Vitousek, P.M., and Chapin, M.C. 2011. Principles of terrestrial ecosystem ecology. 2nd ed. Springer, New York, N.Y.

All figures: The graphs are presented with units, but there are no labels on the y-axes. Please include labels.

> *Thank you for pointing out the mistake. We will add the labels to the y-axis of all graphs in the next version of the manuscript.*

Figure 1a: Why the discrepancy between the "total" and "eddy covariance" in Figure 1? It is unclear from the figure caption, but I think the two curves represent the laboratory derived estimate (total) and the

eddy covariance estimate (eddy covariance) of the same variable? So why is the Total > EC in early season and EC > Total in later season?

> *After reading the comment we realized that our discussion on page 7, L30-38 about this matter is not clear enough. In the updated version of the manuscript we will clarify the discussion to better explain the deviations between the two methods. In the lab we measured potential photosynthesis of plants in their current condition (i.e., under various light levels in 20 °C temperature and optimal moisture, but in the physiological state impacted by moisture conditions in the field) and in the field gas exchange was measured under ambient conditions. The difference in spring between the two methods is likely due to the fact that both vascular plants and Sphagna had high photosynthetic potential (assessed as parameters k and Pmax that we measured in the lab), but were in the field limited by the low temperature. In the constant laboratory temperature of 20 °C this spring time potential was shown as high gross photosynthesis. In the end of the summer, a similar difference in temperature occurred between laboratory and field conditions. At this time, however, photosynthetic potential (again measured as k and Pmax) was low, and therefore the estimates of the two methods were similar. The higher mid-summer eddy covariance-derived GPP in comparison with laboratory measurements likely results from high above 20 °C temperatures concurrently with high PAR levels in the field. We should and will point out more clearly that our study did not take into account temperature dependence of photosynthesis.*

> *Optimally we would have varied temperature as well as PAR while measuring photosynthesis of 19 species in our study site to capture the photosynthesis response to T and PAR over the growing season but unfortunately that was not achievable. Varying two or more factors concurrently has been done in studies focusing on one or few species but in here our main focus was in differences between the species, therefore we were only able to cover potential in one temperature level.*

> *We thank you for the idea to point out in the manuscript that our results indicate ecosystem-level photosynthetic potential may be the highest at different time than ecosystem-level GPP (i.e. photosynthesis "in reality").*

Figure 1d: What does "daily lawn surface water table" represent? I am confused by the word "lawn" (makes me think of a golf course). I suggest removing the smoothing curve and not include any line between dots unless the dots represents continuous daily measurements of water levels (there seems to be a larger data gap around Julian day 210).

> *Thank you for the comment! The term "lawn" is commonly used among peatland ecologists for a surface in peatland having an intermediate water table (in between hummocks and hollows). We did not realize that this term could of course be quite confusing for someone not familiar with such use of that word, especially when it is only defined in the study site description. We will add intermediate peatland surface to the figure legend.*

> *We will gap fill the larger data gap around Julian day 210 using available manual measurements of water table depth.*

Figure 1b: I suggest plotting mean daily air temperature and then present the min and max daily air temperature as a shaded fill behind the mean daily air temperature line.

> *This is a good idea. Will be done.*

Figure 1 (figure caption): Why keeping the laboratory temperature at 20 C during the entire growing season if the mean daily air temperature only reached 20 C during a few days? What is the implication of this approach on the analyses? Can this partly explain the offsets in Fig 1a? We see a large drop in water tables in the field site following Julian day 120. The laboratory measurements tried to keep the temperature and moisture contents constant throughout the season, while the field measurements of air temperature and water table (ie moisture) present rather large variations. How does the limited moisture variability of the laboratory approach affect the overall conclusions stated by the authors? I am worried the authors may have over-stated their findings due to the complex relationships between water, air temperature and photosynthesis found in the field setting, especially considering the deviations in Figure 1a. In combination with Figure 2 (which I assume is based upon laboratory analyses, please clarify in figure text), it looks to me like the vascular plants may have been water-limited (too much water) in their photosynthesis in early season in the field (??)

> *It is a bit complicated that results and discussion are in separate chapters. Now in the results we show the different timing between the two estimates (laboratory and eddy covariance measurements) but we felt that we were fully allowed to discuss this only later, much later than the readers mind. We tried to work towards the discussion by adding daily temperature values as a sub-figure 1b below the comparison in Fig. 1a and by pointing out in Figure legend that lab measurements were conducted under constant 20 °C temperature.*

> *We should and will open up the reasoning behind this choice in the section 2.2. The core of our manuscript is to show the significance of seasonal and interspecific variations in potential photosynthetic light response for the ecosystem level processes. For this we needed to make the measured photosynthetic parameters comparable over the growing season. We could either choose constant temperature and moisture for all samples or measure temperature and moisture response of photosynthesis and the latter was not possible due to the limitations of time. Temperature of 20 °C was selected simply because that is close to the room temperature and also realistic for the field conditions. Photosynthesis measurement devices have a limited capability of regulating the temperature and this temperature was possible to maintain in the laboratory. In our opinion the offsets in Fig. 1a are definitely a result of the constant temperature during the measurements. See also our plan to clarify the discussion section under your question related to Figure 1a.*

> *Although we do agree that the effect of moisture on Sphagnum photosynthesis should be discussed in this paper, we think the constant moisture of the samples during the measurements is not as severe problem as it may seem. The physiological state of mosses is responding to prevailing moisture conditions in the field as shown by Hájek at al. (2009). In the case of Hájek et al. (2009) Sphagnum samples showed physiological differences related to site conditions over two weeks after sampling. Also in this earlier study, samples collected from the field were wetted before measurements. We think the low vascular plant photosynthesis in spring (Fig. 2) is mainly due to low vascular leaf area during that period. However, suffering from excess moisture is an interesting further explanation for this.*

> *Finally, we will clarify all of the figure captions to make it clear, which data is based on laboratory or field analyses.*

Figure 3b. Why the decreasing response of the Sphagnum species throughout the study period? The total seasonal gross photosynthesis is similar between the two methods, but the distribution of those fluxes over the season is rather different between the two methods (laboratory versus eddy covariance). This observation is currently not discussed in the manuscript and I think this is the most interesting piece of the

results. I would like to see the text in the results section to address the seasonal variability that we see in the figures. The results section is currently focusing on the total seasonal values, while the figures show some rather interesting seasonal variations (in time and between methods).

*We think the decreasing photosynthetic potential of Sphagna reflects the decreasing trend of water table over the growing season. Please see below the suggested additional sentences for the discussion to point this out.*

*"The seasonally decreasing Sphagnum $P_G$ is likely to reflect the change in the moisture conditions. Water table depth, which together with precipitation is known to be the most important moderator of Sphagnum photosynthesis (Hayward and Clymo 1983; Backéus 1988; Lindholm 1990; Nijp et al. 2014), decreased at the study site over the growing season (Fig. 1d)."*

*We agree with the reviewer that our discussion regarding the differences between the two methods on page 7, L30-38 should be clarified. In the discussion we will explain more clearly, how the changes in photosynthetic parameters measured at the constant temperature results in different timing of maximum $P_G$ (laboratory measurements) and GPP (eddy covariance measurements). Behind this were most importantly the seasonal changes in light response parameters Pmax (maximum light-saturated photosynthesis) and k (the ability of the plant to use low light levels). See also above our answer to your comment considering Figure 1a.*

*The results regarding seasonal variations of photosynthesis are presented shortly on Page 6, L18-26. We will consider presenting those results more in depth.*

Please refer to specific figures in the discussion.

*We will go through the discussion section and add appropriate references to figures there.*

Page 7, Line 33: The sentence is odd. Remove "when" perhaps?

*You are quite right, removing the word "when" will clarify the sentence.*

The discussion refers to time by naming the month. I suggest all graphs use months instead of Julian day.

*We will add the months as well.*

The discussion or literature does not address the impact on hydrology to the photosynthesis, which, especially for mosses, can have a major impact.

*Thank you for the good point, we should definitely address hydrology discussion. We add the following sentences on page 8, L16: "The seasonally decreasing Sphagnum $P_G$ is likely to reflect the change in the moisture conditions. Water table depth, which together with precipitation is known to be the most important moderator of Sphagnum photosynthesis (Hayward and Clymo 1983; Backéus 1988; Lindholm 1990; Nijp et al. 2014), decreased at the study site over the growing season (Fig. 1d)."*

---

## Referee Comment (RC2) · Anonymous Referee #2 · 28 Sep 2016

General comments

Korrensalo et al. have produced a detailed study with species level in vivo CO2 exchange measurements, produced models of net (PN) and gross photosynthesis (PG) for the measured species, and compare the reconstructed PG's, extrapolated to the ecosystem, with ecosystem gross photosynthesis (GPP) derived from EC tower measurements. The comparison of such different materials are astonishingly good, and show the potential of the methods to contribute to ecosystems models. Not only limited to systems models, the results can be used to test ecological hypotheses as well.

I agree with Referee #1 that one of the most interesting issues revealed is the seasonal gap (June-July) between the PG and GPP, GPP showing higher values. The Supplement contains the estimated parameter values of the species specific, monthly light response functions. Those values somewhat considered in the discussion, but could perhaps be more utilized to inspect the species the live aggregated in the specific microforms with largest changes in LAI and coverage over the season?

The topic is highly relevant to BG, the manuscript offers good data, sound methods, and novel ideas, reaching to conclusions that nicely build on the previous work of the authors I am familiar with. There are some open questions, posed in the specific comments, that may need more work. The manuscript should be publishable after a moderate (major) revision. The issues raised by Referee #1 earlier have already been agreed by the authors in AC1.

Specific comments

1/26-27

The last inference on that "functional diversity may increase the stability of C sink of boreal bogs" comes fro thin air, bacause the concept "functional diversity" was not opened earlier in the abstract. Please modify so that the relationship between the vascular plants and Sphagna that were used in the analyses, and functional diversity becomes clear. Alternatively, use the earlier mentioned study units instead to avoid a hop from species or growth form level to more abstract functional diversity.

2/22-29

One apparent factor may be changes in shading of moss layer due to light extinction under devoloping LAI and coverage. Thus the interaction with the community structure may have imortance.

This is the critical period when the GPP and PG most differ.

Were the irradiation data used logged by the EC? Was any attempts made to estimate light extinction below the changing coverage and LAI of the vasculars? If not, the irradiation seen by the Sphagna living under vasculars may be an overestimate. Could this be significant?

Alm et al (1999) reported on dry-out of mosses that did not recover in terms of photosynthetic capacity after the drought period. Comparison with this may not be valid in all communities?

Language: ... likely to be largely due to ... Uncertainty indicators twise?

7/37-38

Heterotropgic respiration is part of GPP and peaks in field just during the period with highest difference between Pg and GPP. Both WT and temperature control the oxic decomposition. WT is not in the respiration model (Eq. 2). Also perhaps shading of Sphagna. Any comments on these?

Do you refer here to the concept of functional diversity? I think you need be specific on what aspect of diversity is actually in focus here.

9/13-14

The vegetation structure with sparse or dense field layer may also affect the photosynthesis dynamics due to differences in light extinction over the growing season. Any commnets on the basis of S1 table of light response parameters? Another issue is the solar declination that is latitude specific. That could affect the shaded moss assemblages?

---

## Author Response (AR1)

Dear Associate Editor Paul Stoy,

We are pleased to submit to Biogeosciences journal the revised version of the manuscript (bg-2016-265) "Species-specific temporal variation in photosynthesis as a moderator of peatland carbon sequestration" that was under review in the Biogeosciences Discussions forum. We found the comments of the Editor and the reviewers very useful in improving the manuscript. We have incorporated the comments to our manuscript and below, each of them are addressed separately. We would also like to thank for the interactive review process of the journal, since we found this conversational approach very fruitful.

The most substantial comment concerned the seasonal gap between eddy covariance-derived gross primary productivity estimate (GPP) and gross photosynthesis measurements upscaled to the ecosystem level ( $P_G$ ). This issue was a matter of discussion when we were preparing our manuscript and we fully agree with the reviewers that it should be discussed more in depth, which now has been done. In addition to that we have modeled the temperature response of GPP to be able to calculate that for the growing season assuming similar temperature than during our laboratory measurements of photosynthesis. We believe this approach will further clarify the effect of temperature on the difference between GPP and  $P_G$ .

Associate Editor comments to the Author and our response:

Both referees provided insightful reviews and suggested that the manuscript is publishable in Biogeosciences after considering major suggested revisions. Please provide a detailed response to these comments and revise the manuscript accordingly and I will provide the referees the opportunity to decide if the revisions are sufficient to warrant publication in Biogeosciences.

We thank the Editor for offering us the change to resubmit our manuscript. We sincerely believe it was greatly improved after the revisions. Our detailed response to the specific comments is provided below.

Answer to the comments by Anonymous Referee #1

**General comments:**

Korrensalo et al. presents one season of field measurements (eddy covariance), controlled laboratory experiments and modeled results of net and gross photosynthesis rates and/or gross primary production from a boreal bog in southern Finland and emphasizes the species specific contributions and the integration of plot to ecosystem scales. In particular, Korrensalo et al. differentiate between the vascular and bryophyte (moss) contributions, where the latter is oftentimes given a shadow-role in the literature of carbon and energy fluxes. Here, mosses are emphasized to play an important role in the overall wetland ecosystem-level flux. The two approaches in reaching the total system fluxes (eddy covariance and species-specific laboratory experiments) arrive at similar total seasonal fluxes. However, the figures suggest rather large seasonal differences (if I interpret them correctly). I would therefore appreciate increased attention to why that is. In fact, I think this difference is an interesting story (the story?) that emerged. Below are some thoughts that came to me as I reviewed.

We thank the reviewer for the time and effort used to our manuscript and for the thoughtful comments. We agree with the reviewer that the difference between the two methods should be discussed more clearly than it has been done before. We think the seasonal differences between the methods are mainly due to temperature. To clarify

this, we have modeled the temperature response of GPP and simulated seasonal GPP in the same temperature than during our laboratory measurements. We hope the more detailed clarifications below explain the effect of temperature to the reader better than in the previous version of the manuscript.

Please define gross primary production (GPP), net (PN) and gross (PG) photosynthesis so reader who are not regularly working with these terms can follow your manuscript.

We have added the definitions of these terms on P2, L8-12 of the manuscript.

All figures: The graphs are presented with units, but there are no labels on the y-axises. Please include labels.

**We have added the labels in addition to units.**

Figure 1a: Why the discrepancy between the "total" and "eddy covariance" in Figure 1? It is unclear from the figure caption, but I think the two curves represent the laboratory derived estimate (total) and the eddy covariance estimate (eddy covariance) of the same variable? So why is the Total > EC in early season and EC > Total in later season?

First of all, the legend and caption of Figure 1 indeed needed some clarification. We have modified both of them so that it is easier to understand, which estimate came from which method.

After reading the comment we realized that our discussion on page 8, L18-40 about this matter is not clear enough. What we were trying to say there is that in the lab we measured potential photosynthesis of plants in their current conditions, i.e. under various light levels in 20°C and optimal moisture, but in the physiological state impacted by moisture conditions in the field. In the field gas exchange was measured under ambient conditions by the eddy covariance tower. The difference in spring between the two methods is likely due to the fact that both vascular plants and Sphagna had high photosynthetic potential (assessed as parameters k and Pmax that we measured in the lab), but were in the field limited by the low temperature. In the constant laboratory temperature of 20 °C this spring time potential was shown as high gross photosynthesis. In the end of the summer, a similar difference in temperature occurred between laboratory and field conditions. At this time, however, photosynthetic potential (again measured as k and Pmax) was low, and therefore the estimates of the two methods were similar. The higher mid-summer eddy covariancederived GPP in comparison with laboratory measurements is something we cannot fully explain. The lack of vascular plant photosynthesis measurements in July cannot solely explain the deviation, which lasts for two months. We have clarified these points in the discussion on P8, L18-40.

Optimally we would have varied temperature as well as PAR while measuring photosynthesis of the 19 species in our study site to capture the photosynthesis response to T and PAR over the growing season but unfortunately that was not achievable. Varying two or more factors concurrently has been done in studies focusing on one or few species but in here our main focus was in differences between the species, therefore we were only able to cover potential in one temperature level. We added two sentences about this matter on P8, L20-23. To point out more clearly the effect of temperature on the difference between the two methods used, we have added to the manuscript Eq. (4) describing the temperature response of eddy covariance-derived GPP. Using this model, we have simulated the seasonal GPP at constant temperature of 20 °C, same as in the laboratory measurements. This simulated GPP is now presented in the Fig. 1a and we hope the discussion in relation to that (P8, L18-40) clarifies the effect of temperature.

We thank you for the idea to point out in the manuscript that our results indicate that the ecosystem-level photosynthetic potential may peak at a different time than the ecosystem-level GPP (i.e. the "real" photosynthesis).

Figure 1d: What does "daily lawn surface water table" represent? I am confused by the word "lawn" (makes me think of a golf course). I suggest removing the smoothing curve and not include any line between dots unless the dots represents continuous daily measurements of water levels (there seems to be a larger data gap around Julian day 210).

Thank you for the comment! The term "lawn" is commonly used among peatland ecologists for a surface in peatland having an intermediate water table (in between hummocks and hollows). We as non-native speakers did not realize that this term could of course be quite confusing for someone not familiar with such use of that word, especially when it is only defined in the study site description. We added "intermediate peatland surface" to the figure legend.

We used available manual water table measurements to gap fill the larger data gap around Julian day 210. In addition, we removed the smoothing curve from the WT graph (Fig. 1d).

Figure 1b: I suggest plotting mean daily air temperature and then present the min and max daily air temperature as a shaded fill behind the mean daily air temperature line.

*This is a good idea. We added daily minimum and maximum temperature in grey to the graph.*

Figure 1 (figure caption): Why keeping the laboratory temperature at 20 C during the entire growing season if the mean daily air temperature only reached 20 C during a few days? What is the implication of this approach on the analyses? Can this partly explain the offsets in Fig 1a? We see a large drop in water tables in the field site following Julian day 120. The laboratory measurements tried to keep the temperature and moisture contents constant throughout the season, while the field measurements of air temperature and water table (ie moisture) present rather large variations. How does the limited moisture variability of the laboratory approach affect the overall conclusions stated by the authors? I am worried the authors may have over-stated their findings due to the complex relationships between water, air temperature and photosynthesis found in the field setting, especially considering the deviations in Figure 1a. In combination with Figure 2 (which I assume is based upon laboratory analyses, please clarify in figure text), it looks to me like the vascular plants may have been water-limited (too much water) in their photosynthesis in early season in the field (??)

It is a bit complicated that results and discussion are in separate chapters. Now in the results we show the different timing between the two estimates (laboratory and eddy covariance measurements) but we felt that we were fully allowed to discuss this only

later in the discussion section. We have added to Figure 1 seasonal GPP simulated at 20 °C, which we hope will more directly show the effect of temperature on the  $P_G$  and GPP. We also tried to work towards the discussion by adding daily temperature values as a sub-figure 1b below the comparison in Fig. 1a and by pointing out in Figure legend that lab measurements were conducted under constant 20°C temperature.

We opened up a little the reasoning behind the choice to keep the temperature constant during the measurements in the section 2.2. (P3, L36-38). The core of our manuscript is to show the significance of seasonal and interspecific variations in potential photosynthetic light response for the ecosystem level processes. For this we needed to make the measured photosynthetic parameters comparable over the growing season. We could either choose constant temperature and moisture for all samples or measure temperature and moisture response of photosynthesis. Unfortunately, the latter was not possible due to the limitations of time. Temperature of 20 °C was selected simply because that is close to the room temperature and also realistic for the field conditions. Photosynthesis measurement devices have a limited capability of regulating the temperature and this temperature was possible to maintain in the laboratory. In our opinion the offsets in Fig. 1a are definitely a result of the constant temperature during the measurements. See also our revisions under your question related to Figure 1a, which intend to clarify the discussion section.

We do agree that the effect of moisture on Sphagnum photosynthesis should be better discussed in this manuscript, which has now been done on P9, L15-18. However, we think the constant moisture of the samples during the measurements is not as severe problem as it may seem. The physiological state of mosses is responding to prevailing moisture conditions in the field as shown by Hájek at al. (2009): Sphagnum samples showed physiological differences related to site conditions over two weeks after sampling. Also in this earlier study, samples collected from the field were wetted before measurements. We think the low vascular plant photosynthesis in spring (Fig. 2) is mainly due to low vascular leaf area during that period. However, suffering from excess moisture is an interesting further explanation for this.

Finally, we have clarified all of the figure captions to make it clear, which data is based on laboratory or field measurements.

Figure 3b. Why the decreasing response of the Sphagnum species throughout the study period? The total seasonal gross photosynthesis is similar between the two methods, but the distribution of those fluxes over the season is rather different between the two methods (laboratory versus eddy covariance). This observation is currently not discussed in the manuscript and I think this is the most interesting piece of the results. I would like to see the text in the results section to address the seasonal variability that we see in the figures. The results section is currently focusing on the total seasonal values, while the figures show some rather interesting seasonal variations (in time and between methods).

We think that the decreasing photosynthetic potential of Sphagna reflects the decreasing trend of water table over the growing season. Please see on P9, L15-18 the suggested additional sentences for the discussion to point this out.

We agree with the reviewer that our discussion regarding the differences between the two methods should be clarified. Please see section 4.1, which now has been modified to meet this demand.

The results regarding seasonal variations of photosynthesis have now been presented more in depth in section 3.1.

Please refer to specific figures in the discussion.

We went through the discussion section and added appropriate references to figures.

Page 7, Line 33: The sentence is odd. Remove "when" perhaps?

You are quite right, removing the word "when" clarified the sentence.

The discussion refers to time by naming the month. I suggest all graphs use months instead of Julian day.

We have now added the months as well.

The discussion or literature does not address the impact on hydrology to the photosynthesis, which, especially for mosses, can have a major impact.

Thank you for the good point, we should definitely address hydrology in discussion. We have added some discussion about this matter on P9, L15-18.

**Answer to the comments by Anonymous Referee #2**

General comments

Korrensalo et al. have produced a detailed study with species level in vivo CO2 exchange measurements, produced models of net (PN) and gross photosynthesis (PG) for the measured species, and compare the reconstructed PG's, extrapolated to the ecosystem, with ecosystem gross photosynthesis (GPP) derived from EC tower measurements. The comparison of such different materials are astonishingly good, and show the potential of the methods to contribute to ecosystems models. Not only limited to systems models, the results can be used to test ecological hypotheses as well. I agree with Referee #1 that one of the most interesting issues revealed is the seasonal gap (June-July) between the PG and GPP, GPP showing higher values. The Supplement contains the estimated parameter values of the species specific, monthly light response functions. Those values somewhat considered in the discussion, but could perhaps be more utilized to inspect the species the live aggregated in the specific microforms with largest changes in LAI and coverage over the season? The topic is highly relevant to BG, the manuscript offers good data, sound methods, and novel ideas, reaching to conclusions that nicely build on the previous work of the authors I am familiar with. There are some open questions, posed in the specific comments, that may need more work. The manuscript should be publishable after a moderate (major) revision. The issues raised by Referee #1 earlier have already been agreed by the authors in AC1.

> We would like to thank the reviewer for the overall positive comment on our manuscript and for making the effort to read the comments of the previous reviewer as well as our interactive comment on that. We fully agree with the general comment that the values presented in the supplementary data should be discussed more in depth. This has been done in another manuscript, which was recently accepted to another journal, but has not yet been published. In addition, we are currently preparing a manuscript where the differences in  $CO_2$  balance among vegetation communities

growing on different microforms are studied. We have addressed the comments by the reviewer separately below.

**Specific comments**

**1/26-27**

The last inference on that "functional diversity may increase the stability of C sink of boreal bogs" comes fro thin air, bacause the concept "functional diversity" was not opened earlier in the abstract. Please modify so that the relationship between the vascular plants and Sphagna that were used in the analyses, and functional diversity becomes clear. Alternatively, use the earlier mentioned study units instead to avoid a hop from species or growth form level to more abstract functional diversity.

You are right, we clearly lost some essential information when trying to reduce the word count of the abstract. Please see the new version of the abstract, where functional diversity is better defined.

**2/22-29**

One apparent factor may be changes in shading of moss layer due to light extinction under devoloping LAI and coverage. Thus the interaction with the community structure may have imortance.

We understand this concern well, but, knowing the site very well, we do not think the shading of the moss layer plays a large role in Siikaneva site. We would like to point out the low maximum LAI at the site (Fig. 1c) and demonstrate the very sparse vascular layer with the photograph below. On P8, L4-7 we added a sentence: "Although the shading of the moss layer by vascular plants may figure as a potential error source of  $P_G$  upscaled with PPFD measured above the vegetation, it is not likely to be caused by the sparse cover of vascular plants at the site (Supplementary information, Fig. S3) with low seasonal maximum LAI (Fig. 1c)." We have also added the site photograph to the supplementary information.

This is the critical period when the GPP and PG most differ.

Thank you for bringing this up. We have now discussed the effect of lacking measurements on P8, L29-32.

**5/4**

Were the irradiation data used logged by the EC? Was any attempts made to estimate light extinction below the changing coverage and LAI of the vasculars? If not, the irradiation seen by the Sphagna living under vasculars may be an overestimate. Could this be significant?

On P5, L9-10 it is told that the source of the light data is a measurement station close to the site. To clarify the height were light was measured, we also added word "above-canopy" to those lines. Because of the sparse vascular vegetation at our site, we did not attempt to estimate the difference in light level above and below the canopy. We have assumed that shading has only a negligible effect on the moss photosynthesis estimate, but of course, it would have been a good idea to quantify this. Please see also our answer to the comment regarding P2, L22-29 and the attached photograph where the sparseness of vascular vegetation is demonstrated.

**7/25**

Alm et al (1999) reported on dry-out of mosses that did not recover in terms of photosynthetic capacity after the drought period. Comparison with this may not be valid in all communities?

This is true, and we have now noted that on P8, L12-13. In this study we did not compare the vegetation communities, so we would like to limit the comparison with the article by Alm et al. (1999) to the ecosystem-level estimate of cumulative growing season gross photosynthesis. However, in our future work we will concentrate on the differences among plant communities in  $CO_2$  balance.

**7/31**

Language: ...likely to be largely due to ... Uncertainty indicators twise?

Thank you for noticing the mistake. The sentence is now rewritten: "The shape of  $P_G$  and GPP development differed over the growing season, especially at the beginning of the summer, which is largely due to the constant temperature of 20 °C in our laboratory measurements (Fig. 1b)."

**7/37-38**

Heterotropgic respiration is part of GPP and peaks in field just during the period with highest difference between Pg and GPP. Both WT and temperature control the oxic decomposition. WT is not in the respiration model (Eq. 2). Also perhaps shading of Sphagna. Any comments on these?

The residuals of Eq. (2) did not correlate with WT level, and we therefore did not include WT in the model. However, this can be partly due to the limited WT range of the nighttime data to which the respiration model was fitted. If requested by the reviewer, we are happy to add a sentence about this to the Method section.

Above we visualized the low coverage of vascular plants with a photograph (found also in Supplementary information, Fig. S3). Because of this, we do not think that the shading of Sphagna plays a large role at our site.

Do you refer here to the concept of functional diversity? I think you need be specific on what aspect of diversity is actually in focus here.

Thank you for pointing this out. We have now clarified the paragraph and hope the changes on P10, L11-15 make it sufficiently specific.

**9/13-14**

The vegetation structure with sparse or dense field layer may also affect the photosynthesis dynamics due to differences in light extinction over the growing season. Any commnets on the basis of S1 table of light response parameters? Another issue is the solar declination that is latitude specific. That could affect the shaded moss assemblages?

We agree with the reviewer, that in many ecosystems shading of the mosses by vascular plants would indeed be an important issue, possible also having an effect of the seasonal changes in light response parameters of photosynthesis. We added above a photograph, which we think shows that our site is an exception to that because of the low coverage of vascular plants, which also can be seen as low maximum LAI (Fig. 1c).

**9/9**

**Species-specific temporal variation in photosynthesis as a moderator of peatland carbon sequestration**

Aino Korrensalo1, Tomáš Hájek2, Pavel Alekseychik3, Janne Rinne4, Timo Vesala3,5, Lauri Mehtätalo6, Ivan Mammarellac, Eeva-Stiina Tuittila1

5 1School of Forest Sciences, University of Eastern Finland, Joensuu, Finland

2 Faculty of Science, University of South Bohemia, České Budějovice, Czech Republic

3 Dept. of Physics, University of Helsinki, Helsinki, Finland

4 Dept. of Physical Geography and Ecosystem Science, Lund University, Lund, Sweden

5 Dept. of Forest Sciences, University of Helsinki, Helsinki, Finland

10

15

25

6 School of Computing, University of Eastern Finland, Joensuu, Finland

**Correspondence to: aino.korrensalo@uef.fi**

Abstract. In boreal bogs plant species are low in number, but they differ greatly in their growth forms and photosynthetic properties. We assessed how ecosystem carbon (C) sink dynamics were affected by seasonal variations in photosynthetic rate and leaf area of different species. Photosynthetic properties (light-response parameters), leaf area development and areal cover (abundance) of the species were used to quantify species - specific net and gross photosynthesis rates ( $P_N$  and  $P_G$ , respectively), which were summed to express ecosystem-level  $P_N$  and  $P_G$ . The ecosystem-level  $P_G$  was compared with a gross primary production (GPP) estimate derived from eddy covariance measurements (EC).

Species areal covers rather than differences in photosynthetic properties, determined the species with the highest
 PG of both vascular plants and *Sphagna*. Species-specific contributions to the ecosystem PG varied over the growing season, which, in turn, determined the seasonal variation in ecosystem PG. The upscaled growing-season PG estimate, 230 g C m-2, agreed well with the GPP estimated by the EC, 243 g C m-2.

Sphagna were superior to vascular plants in ecosystem-level PG throughout the growing season2 but had a lower PN. PN results indicated that areal cover of the species2 together with their differences in photosynthetic parameters3 
[revised manuscript text omitted]
 aAir temperature was at constantset to 20 °C, the flow rate atto 600 µmol s-1 and the CO2 concentration in the incoming air atto 400 ppm

to be able to compare the seasonal changes in photosynthetic potential among species. The relative humidity inside the cuvette was kept at 50% for the vascular plants and below 90% for the Sphagna. The measured Net

photosynthesis ( $P_N$  value) of each sample at the three light levels was expressed per photosynthesizing leaf area (mg CO2 m-2 (LA) h-1), which was the leaf area inside the cuvette measured with a scanner for vascular plants and assumed to be the cuvette area for *Sphagna*. Two of the species, namely *Rhynchospora alba* and *Rubus chamaemorus*, were not yet of measurable size in May; *R. alba* had already mostly senesced in September and therefore were not measured in those months. Altogether, the data consisted of 720 measurements.

**2.3 Net photosynthesis model**

5

10

15

20

25

To obtain a species-wise flux reconstruction of  $P_N$  and  $P_G$ , we fitted a nonlinear mixed-effects model separately for each combination of species and month. Mixed-effects modeling approach allowed us to take into account the variation between samples, of which each was measured at three light levels. We used the hyperbolic light saturation curve of  $P_N$  (Larcher, 2003) (Eq. (1)):

$$PN_{si} = R_s + \frac{Pmax_s PPFDsi}{k_s + PPFDsi} + e_{si}$$
(1)

where  $PN_{si}$  is the observed net CO2 exchange (mg CO2 m-2 (LA) h-1) and *PPFDsi* 
[revised manuscript text omitted]
(CO2) m-2 h-1, *k* was 170.3 µmol m-2 s-1 was 0.1, *Teot* was 22.6 °C and *Teot* was 20.9 °C.

Cumulative growing season gross photosynthesis (PG) upscaled to the ecosystem level using the separate light response curves for species and months (Eq. (1)) was 230 g C m-2 (Julian days 121–273). Daily PG estimates were higher than GPP values from the EC tower in spring and, lower in the middle of the summer and quite similar in the autumn (Fig. 1a). The GPP simulated at 20 °C, the same temperature as during the laboratory measurements, was similar than upscaled PG in spring but closer to the measured GPP in the middle of the summer (Fig. 1a). In the autumn, all of the three estimates showed rather similar levels (Fig. 1a).

(4)

| -{ | Formatted: Subscript |
|----|----------------------|
| 1  | Formatted: Subscript |
| Υ  | Formatted: Subscript |

| Formatted: Font: Italic               |
|---------------------------------------|
| Formatted: Font: Italic               |
| Formatted: Not Superscript/ Subscript |
| Formatted: Font: Italic               |
| Formatted: Font: Italic               |
| Formatted: Font: Italic, Subscript    |
| Formatted: Font: Italic               |
| Formatted: Font: Italic, Subscript    |
| Formatted: Subscript                  |

*Sphagna* at the study site had higher cumulative growing season  $P_G$  value (138 g C m-2) than vascular plants (92 g C m-2). *Sphagna* had higher daily  $P_G$  than vascular plants in spring and autumn\_but were almost at the same level in the middle of the summer (Fig. 2). A small increase in *Sphagnum* photosynthesis was observed during May (Fig. 2 and 3b) due to increment of daily PPFD towards midsummer. Otherwise, *Sphagnum*  $P_G$  decreased steadily over the growing season (Fig. 2). Seasonal changes in vascular  $P_G$  showed similar patterns than vascular LAI

5 over the growing season (Fig. 2). Seasonal changes in vascular  $P_{G}$  showed similar patterns than vascular LAI development, although the maximum  $P_{G}$  was reached slightly earlier in the season than maximum LAI (Fig. 1a, Ic and 2).

The three vascular plant species having the highest  $P_G$  in the ecosystem were *C. vulgaris*, *R. alba* and *A. polifolia*. *A. polifolia* was the most productive species in May and September, *R. alba* in June and July and *C. vulgaris* in

10 August (Fig. 3a and 4a). With 13% cover altogether (Table 1), they formed 22% of the seasonal ecosystem PG and 56% of the vascular plant PG (Fig. 4). The three *Sphagnum* species with highest PG at the ecosystem level were *S. papillosum*, *S. fuscum* and *S. rubellum* (Fig. 3b and 4b). As with all of the *Sphagnum* species, their PG per ground area decreased steadily over the growing season (Fig. 2 and 3b). With 42% cover altogether (Table 1), they formed 40% of the seasonal ecosystem PG, 67% of the PG of *Sphagnum* mosses (Fig. 4).

**15 3.2 Cumulative growing season net photosynthesis**

The aboveground vegetation of the study site was a carbon sink of 77 g C m-2 over the growing season as estimated by  $P_N$  value upscaled to ecosystem level using the species- and month-wise light response curves.  $P_N$  results for *Sphagna* and vascular plants were reversed in comparison to  $P_G$  estimates;  $P_N$  of *Sphagna* was 20 g C m-2 and vascular  $P_N$  was 57 g C m-2.

20 The same vascular plant species had the highest growing season  $P_N$  and  $P_G$ ; *R. alba, C. vulgaris* and *A. polifolia* had the highest  $P_N$  estimates of  $15.\underline{19}, \underline{9.1}$  and  $8\underline{.4}$  g C m-2, respectively (Table 1). These three species made up 57% of the total vascular  $P_N$  and 42% of the whole ecosystem-level  $P_N$ .

S. fuscum, S. papillosum, and S. majus had the highest seasonal PN of Sphagnum species 7.4, 6.8.7 and 2.83 g C m-2, respectively (Table 1). The PN of these three species was 85% of the total Sphagnum PN and 22% of the seasonal ecosystem PN. Although having-a one of the highest coverage and PG, S. rubellum was not among the three most productive species in terms of PN.

**4 Discussion**

**4.1 Comparison of upscaled gross photosynthesis values with eddy covariance gross primary production vity estimates**

Accounting for the differences in photosynthetic parameters between species and between phases of the growing season appeared to accurately estimate ecosystem PG when upscaling species level measurements. *Sphagnum* mosses especially showed a large seasonal variation in their photosynthetic light response, which could be accounted for in this upscaling approach. The similarity of the PG estimates calculated with species-wise and monthly light response curves and GPP estimates derived from eddy covarianceEC measurements (Fig. 1a), adds
 credibility to the methods used and indicates that the photosynthetic parameters measured under laboratory conditions are comparable with field measurements. Both methods carry their error sources. Annual CO2 flux

[revised manuscript text omitted]

---

## Author Response (AR2)

Dear Associate Editor Paul Stoy,

We are pleased to submit to Biogeosciences journal the revised version of the manuscript (bg-2016-265) "Species-specific temporal variation in photosynthesis as a moderator of peatland carbon sequestration" that was under review for the second time. We would like to thank the Editor and the reviewers for the valuable comments, which now have been incorporated to our manuscript. Below, each of the comments are addressed separately.

The largest revision suggested was to include uncertainty estimates to our eddy covariance –derived GPP and upscaled gross photosynthesis ($P_G$) estimate. We do agree with the reviewer that the uncertainty estimates would be very useful when comparing GPP and $P_G$, and the uncertainty estimate of GPP has now been added to our manuscript. Unfortunately, there are problems in calculating an uncertainty estimate for $P_G$, which was the reason why we originally decided to leave it out of our manuscript. We have below explained our reasoning behind that choice. Naturally, if requested by the Editor, we are prepared to try to find a way to calculate as good uncertainty estimate as possible, even though it would require quite much more modelling effort.

We would also like to point out that the order of second and third author has been switched in this version of the manuscript. This has been done with the acceptance of them both after reconsidering the weight of different methods for this work.

Associate Editor comments to the Author and our response:

Both referees find the manuscript publishable in Biogeosciences after minor revisions and I agree with their assessment. Please consider the remaining referee comments in a brief letter and improve the manuscript accordingly, and I will be happy to recommend it be published in Biogeosciences if the responses adequately account for the recommendations of the referees.

Sincerely,
Paul C. Stoy

> *We thank the Editor for offering us the change to resubmit our manuscript. Our detailed response to the specific comments is provided below.*

Answer to the comments by Anonymous Referee #1

The authors have adequately addressed the details from earlier revision and the manuscript is nearly ready for publication in BG. I have only a few minor suggestions, mainly for improving the readability.

Page 3, line 2
Since climate is apparently warming it would be reasonable to give the actual 30 year period used for the 30 year averages.

> *We have added the years used for the 30 year averages.*

Page 8, line 14
Use plural: The shapes pf PG and GPP development…

> *We have now used plural in this sentence.*

Page 8, lines 16-19

Two complicated sentences. Could be rephrased, e.g.
The constant temperature, maintained in laboratory, allowed us to investigate how the changes in species photosynthetic parameters were affected by seasonal changes in moisture conditions in the field. Since measuring of species-specific temperature responses of PG was unachievable due to the large number of species, we instead chose to model the temperature dependence of EC-derived GPP (Eq. (4), Fig. 1a).

> *We have now reformulated the two sentences according to the suggestions of the reviewer.*

Page 9, lines 3-4
A complicated sentence. Could be rephrased, e.g.
No inter-species differences in photosynthetic properties, either within vascular plants or Sphagna, could change this order.

> *We have reformulated the sentence as suggested by the reviewer.*

Page 10, lines 27-29
Sentence "The diversity of vegetation… (Fig. 3a and b)."
may repeat what was already said on line 19-21. Please check.

> *This is true, thank you for noticing. We have modified the text to avoid repetition.*

Answer to the comments by Anonymous Referee #2

I find the manuscript to be interesting and well written, but I struggled with the lack of uncertainty analyses about the eddy covariance and upscaled chamber-based observations, particularly when the model in equation 1 explicitly includes an uncertainty term. The discussion of potential sources of bias in the Discussion was nice, but uncertainty estimates should still be published. On the point of potential biases, I also find it important to include at least a mention to the notion that eddy covariance-based GPP estimates are likely overestimates due to recent findings that the Kok effect is likely larger than previously assumed (Wehr et al., 2016).

Wehr, R., Munger, J.W., McManus, J.B., Nelson, D.D., Zahniser, M.S., Davidson, E.A., Wofsy, S.C., Saleska, S.R., 2016. Seasonality of temperate forest photosynthesis and daytime respiration. Nature 534, 680–683.

> *We would like to thank the reviewer for the positive statement regarding our manuscript. We have now added the suggested reference to our manuscript on P9, L11-13 (page and row numbers refer to the revised version).*
>
> *As stated above, we fully agree with the reviewer that the manuscript would benefit from including the uncertainty estimates. We have added a 95% confidence interval to our eddy covariance –derived GPP, P7, L26 and an explanation how it was calculated on P7, L10-13.*
>
> *To calculate the uncertainty estimate of photosynthesis, we would have to take into account 1) the estimation errors of the fixed parameters, 2) the random effects of our mixed-effects models and 3) residual variance. For the second point we would need to have quantified the correlation of the random effects over time and between species. A proper model for such purpose would be a nonlinear models system that models correlated random effects and residual errors over time. However, model fitting procedures for such model are not available in the standard statistical software.*

*These problems are now explained in the manuscript on P5, L18-24. We tried to fit a less complicated model which assumed hierarchical nested random effects but we could not fit such a model due to estimation problems in the model fit. Modeling temporal and between-species correlation of random effects would be especially important for the uncertainty calculation, since almost all of our 87 models show that the random effects of the models account for several times more variation than the residual variation (Supplementary table 1). On the other hand, the large variation in the random effects compared with the residual variation supports our choice to use mixed-effects models, which takes into account the random variation between samples. Although the current modeling structure makes it difficult to calculate only one uncertainty estimate for the whole growing season, we sincerely believe this was the most accurate approach to estimate the light response parameters.*

*To put it short, calculating the uncertainty estimate of upscaled photosynthesis is quite complicated, and we are afraid such an estimate would very likely be erroneous. Therefore, we would prefer staying in our decision not to present it in the manuscript. Currently, the uncertainties of the 87 models are presented in the Supplement 1. However, we are willing to do our best to find a way to calculate an uncertainty estimate for the whole growing season, if requested by the Editor.*

[revised manuscript text omitted]